# Modeling Hierarchical Thinking in Large Reasoning Models

**G M Shahariar** [* 1]  **Erfan Shayegani** [* 1]  **Ali Nazari** [2]  **Nael Abu-Ghazaleh** [1]

## Abstract

Large Reasoning Models (LRMs) solve complex tasks by generating long Chain-of-Thought (CoT) sequences; however, the emergent dynamics governing reasoning trajectories are not well understood and can lead to inconsistencies and reasoning pathologies. In this work, we propose to approximate LRM's emerging hierarchical reasoning dynamics as a trajectory within a Finite State Machine (FSM) transitioning among six abstract cognitive states. We demonstrate that these states and transitions can be captured in the latent state of the model. We believe that this representation can have different applications in the interpretability and optimization of LRM models. For example, by analyzing the topology of these transitions, we identify statistical shifts in reasoning strategies that help identify effective reasoning chains from those that fail. To illustrate these potential advantages, we propose $Q$-Value guided steering, a training-free inference-time control method that treats reasoning as a planning problem. We estimate the long-horizon utility of state transitions and apply sparse, orthogonal activation steering at sentence boundaries to align the CoT generation with optimal reasoning policies. Experiments across four benchmarks (AIME25, MATH-500, GSM8k, and GPQA Diamond) using three state-of-the-art open reasoning models demonstrate that $Q$-Value steering policy achieves significant performance gains with "surgical" efficiency, often requiring $25\times$ fewer interventions than greedy and weighted baselines, which suggests that reasoning can be effectively controlled by guiding high-level cognitive dynamics rather than micromanaging token generation. Code is available at: https://github.com/shahariar-shibli/CoT-FSM.

*Equal contribution  [1]University of California, Riverside  [2]Independent Researcher. Correspondence to: G M Shahariar <gshah010@ucr.edu>, Erfan Shayegani <sshay004@ucr.edu>.

*Proceedings of the 43rd International Conference on Machine Learning*, Seoul, South Korea. PMLR 306, 2026. Copyright 2026 by the author(s).

## 1. Introduction

Large Language Models (LLMs), when trained to elicit Chain-of-Thought (CoT) reasoning, demonstrate impressive reasoning capabilities by generating step-by-step solutions to complex problems (Kojima et al., 2022; Wei et al., 2022). The resulting *Large Reasoning Models* (LRMs) often exhibit behavior that resembles human cognition: They generate hierarchical reasoning strategies, effectively "thinking out loud" before reaching a conclusion, which leads to better-reasoned responses even for complex instructions. Despite their empirical success, the internal dynamics governing these emergent reasoning processes remain poorly understood. A single reasoning trajectory may span thousands of tokens and involve different cognitive phases, and it is unclear whether these emergent trajectories are prone to inefficient reasoning and other pathologies. This presents a fundamental challenge: without a structured map of *how* a model moves between cognitive phases, it is hard to diagnose *why* a trajectory succeeds or fails.

Recent works have addressed this challenge mainly through *local adjustments* or *behavior-specific* lenses. Chen et al. (2025a) improved reasoning efficiency by suppressing "unproductive" thought patterns via representation engineering, but they largely treated overthinking as a coarse binary phenomenon (execution vs. reflection/transition). Venhoff et al. (2025) identified that some reasoning behaviors in LRMs are mediated by linear directions in the model's activation space and can be modulated causally at inference time. Although these approaches are effective for local behavioral adjustments or pruning, they lack a *global* hierarchical view of how cognitive modes evolve throughout a reasoning trajectory. They also fail to answer a key control question: *given where the model currently is in its reasoning, what is the best next cognitive move to maximize correctness?* In practice, existing works tend to (i) target a single behavior category at a time, (ii) rely on heuristics such as suppressing redundancy, or (iii) demonstrate controllability without explicitly linking *sequences of cognitive shifts* to downstream task success. This leaves a gap between interpretability (identifying steerable behaviors) and actionable control (deciding *when* and *where* to intervene along a trajectory).

In this work, to bridge this gap, we analyze the hierarchical thinking in LRMs through the lens of a Finite State Machine

(FSM). We propose a rigorous analytical framework where the CoT reasoning is abstracted into a discrete sequence of six high-level states: *Initialization, Deduction, Augmentation Strategy, Uncertainty Estimation, Backtracking,* and *Final Conclusion*. Projecting free-form CoT into this state space yields a compact state transition graph whose transition structure can be estimated at scale and compared between correct vs. incorrect solutions. This allows us to move beyond per-behavior trajectory control and instead quantify *which transitions* (e.g., moving from uncertainty to backtracking) are statistically associated with success and which transitions correlate with failure.

We show that our FSM abstraction is not only an analysis tool, but also a controllable interface for training-free inference-time intervention. By training-free, we refer to no weight updates during inference to the reasoning model weights. We view the FSM abstraction as a planning environment and derive a long-horizon CoT control policy using $Q$-value iteration, enabling *sparse* and *strategic* interventions through activation steering to nudge the model toward outcome-aligned transitions determined by the policy. Across multiple open reasoning models (*Qwen3-4B-Thinking* (Yang et al., 2025), *Phi-4-reasoning* (Abdin et al., 2025), *gpt-oss-20b* (Agarwal et al., 2025)) and challenging benchmarks including AIME25 (MAA Committee, 2025), MATH-500 (Hendrycks et al., 2021), GPQA Diamond (Rein et al., 2023) and GSM8k (Cobbe et al., 2021), our planning-aware approach ($Q$-Value steering) yields consistent performance gains while requiring far fewer interventions than greedy or weighted baselines, suggesting that reasoning can be effectively controlled by guiding high-level cognitive dynamics.

Throughout this work, we use cognitive terminology — such as 'reasoning states,' 'uncertainty,' and 'backtracking' — as functional descriptors for observable patterns in CoT trajectories, not as claims about the presence of human-like cognition. Whether these emergent behavioral patterns reflect genuine metacognitive processes analogous to human thinking remains an open question; our framework models their structure and demonstrates that intervening on them improves reasoning outcomes. In summary, our main contributions are as follows:

- **FSM Abstraction of Reasoning.** We introduce a formal state-space taxonomy for LRM reasoning, decomposing CoT into a discrete trajectory of six functional cognitive states.

- **Transition Dynamics Analysis:** We define *Transition Advantage Matrix*, a quantitative metric that identifies which cognitive shifts are statistically associated with correct vs. incorrect outcomes.

- **Planning-aware Steering Policy.** We propose a training-free inference-time control framework that treats reason-

ing as a planning problem; via $Q$-value iteration, we estimate long-horizon utilities and use them to gate and guide activation steering at sentence boundaries.

- **Efficiency and Performance:** We demonstrate that FSM-guided steering significantly improves performance on complex reasoning tasks (e.g., +13% accuracy on AIME25 for *gpt-oss-20b* with low reasoning effort) while substantially reducing the number of required interventions compared to stronger heuristic baselines.

## 2. Finite State Abstraction of LLM Reasoning

We formalize the reasoning process of an LRM into a rigorous analytical framework (Figure 2 (a)) by modeling reasoning as a trajectory through a discrete state space and representing it as a Finite State Machine (FSM). FSM is a memoryless state machine, providing a simple model for capturing the reasoning progression. We recognize that more complex models, those that admit model state/memory, may eventually capture reasoning dynamics more accurately, as transition decisions may depend on the reasoning state/memory, but leave such explorations to future research.

### 2.1. Reasoning as a Sentence-Level Trajectory

Let $\mathcal{M}$ be a Large Reasoning Model (LRM). Given a prompt $\mathcal{P}$, the model generates a Chain-of-Thought, $\mathcal{C} = (x_1, x_2, \ldots, x_N)$, where $x_i$ represents the $i$-th token in the CoT. However, tokens are often too granular to capture high-level reasoning steps because a single logical inference or hypothesis generation may span dozens or hundreds of tokens. Therefore, following prior works (Bogdan et al., 2025; Venhoff et al., 2025), we decompose $\mathcal{C}$ into a sequence of sentences $\mathcal{S} = (s_1, s_2, \ldots, s_K)$.

### 2.2. Discrete Abstract Reasoning States

**State Taxonomy.** We define a set of six high-level reasoning states that commonly appear in CoT trajectories: $\mathcal{Q} = \{$*init*, *deduce*, *augment*, *uncertain*, *backtrack*, *closure*$\}$ as shown in Figure 1. The taxonomy may vary by domain, but we found it sufficient for modeling chain-of-thoughts on complex mathematics and scientific knowledge QA tasks in our experiments.

Modeling hierarchical reasoning into abstract states connects with classical models from human problem-solving theory, such as the well-known four-step framework by Polya (1945) – *understand the problem, devise a plan, carry out the plan,* and *look back.* This is perhaps not surprising since LRMs are trained on chain-of-thought examples obtained from human reasoning. In our FSM formulation, *initialization* roughly corresponds to understanding the problem, *augmentation strategy* relates to devising a plan, *deduction* is carrying out the plan, and *backtracking* corresponds

⏳ **Initialization (init):** Restating or clarifying the problem, or setting up the proposed approach.

🔍 **Deduction (deduce):** Drawing inferences or doing step-by-step logical reasoning based on current information.

◑ **Augmentation Strategy (augment):** Introducing new information or strategies; e.g., recalling facts, breaking the problem into sub-problems, or exploring alternative solution paths.

☹ **Uncertainty Estimation (uncertain):** Explicitly expressing doubt, assessing confidence, or identifying potential errors, often a prelude to revising the approach.

↺ **Backtracking (backtrack):** Revisiting and revising earlier steps/ assumptions; effectively rewinding to try a different approach after recognizing a mistake or dead-end.

⊘ **Final Conclusion (closure):** Presenting a final answer or a conclusive summary of the solution, typically the end of the reasoning trajectory.

*Figure 1.* Six high-level reasoning states used to model Chain-of-Thought trajectories.

to looking back and revising as needed. We extend Polya's framework with two additional categories observed in LRM behavior: *uncertainty estimation* and *final conclusion*.

Besides Polya's framework, our state taxonomy connects to another well-established cognitive framework – Schoenfeld's Episode Theory (Schoenfeld, 2014), which identifies six foundational episodes (*read, analyze, plan, implement, explore, verify*) as necessary and sufficient for complex problem-solving. Our taxonomy, based on empirical observations of the chains of thought, also adapts this classical architecture to the context of LRMs by mapping *initialization* to read/analyze, *deduction* to implementation, *augmentation* to plan/explore. *Uncertainty estimation* and *backtracking* states operationalize metacognitive monitoring and control functions (Nelson, 1990), critical for capturing the non-linear, self-corrective behaviors unique to LRMs which maps to verify. We provide a brief description of the state taxonomy in Appendix A.

**State Assignment.** We define a state assignment function $\phi : \mathcal{S} \to \mathcal{Q}$ that maps each sentence $s_k$ to a state $q_k \in \mathcal{Q}$. In our framework, we implement this function via automated annotation using a foundation model (*GPT-4o-mini*), since GPT-variants have shown to achieve high annotator agreement for functional labeling in prior works (Jahan et al., 2024; Liyanage et al., 2024). Therefore, applying $\phi$ on a CoT $\mathcal{C}$ results in a discrete state reasoning trajectory: $\mathcal{T} = (q_1, q_2, \ldots, q_K), \quad q_k \in \mathcal{Q}$.

### 2.3. FSM Formalization

We define the CoT reasoning process as a Finite State Machine represented by the 5-tuple: $M = (Q, \Sigma, \delta, q_0, F)$, where $Q$ is the set of six abstract states previously defined, $\Sigma = \{\varepsilon\}$ is a trivial input alphabet containing only the empty symbol (as state transitions occur without any external input), $\delta : Q \times \Sigma \to Q$ is the transition function that maps each state to a set of possible successor states in the reasoning trajectory, $q_0 \in Q \setminus \{closure\}$ is the initial state (typically *init*, however, any non-terminal state may serve as a starting point), and $F = \{closure\}$ is the singleton set containing the final accepting state.

### 2.4. Transition Dynamics

**Transition Graph.** The FSM formalization allows us to view reasoning not as a linear sequence of tokens, but as a traversal through a functional graph. The nodes correspond to the six abstract states, and the edges represent the transitions between these states. In a traditional FSM, transitions are often triggered by external inputs. In our framework, the transitions are endogenous - they are generated by the model's own previous outputs. The model reads its own previous thoughts (past states) and decides to move to the next thought (next possible state).

**Transition Probability.** For every directed transition $(q_i \to q_j)$ in the transition graph, we can count its total occurrences across all reasoning trajectories on a dataset and compute the normalized transition probability as

$$P(q_j \mid q_i) = \frac{\sum_{n=1}^{N} C_{\mathcal{R}_n}(q_i \to q_j)}{\sum_{k} \sum_{n=1}^{N} C_{\mathcal{R}_n}(q_i \to q_k)},$$

where $C_{\mathcal{R}_n}(q_i \to q_j)$ denotes the number of times the transition $q_i \to q_j$ appears in a trajectory $\mathcal{R}_n$. This captures the global transition dynamics of reasoning states within a dataset.

**Outcome-Conditioned Transition Matrices.** We can view the transition dynamics as conditioned on the eventual outcome of the reasoning trajectory, denoted by the binary variable $O \in \{Success, Failure\}$. We define two conditional transition matrices:

$$T_{ij}^{(correct)} = P(q_{t+1} = j \mid q_t = i, O = Success)$$

$$T_{ij}^{(incorrect)} = P(q_{t+1} = j \mid q_t = i, O = Failure)$$

These matrices capture two distinct policies: one corresponding to successful reasoning and the other to failed trajectories. Specifically, $T_{ij}^{(correct)}$ and $T_{ij}^{(incorrect)}$ represent the probabilities of transitioning from state $i$ to $j$ under correct and incorrect outcomes, respectively.

**Transition Advantage Matrix** $(R)$**.** To identify the transitions that distinguish success from failure, we define an

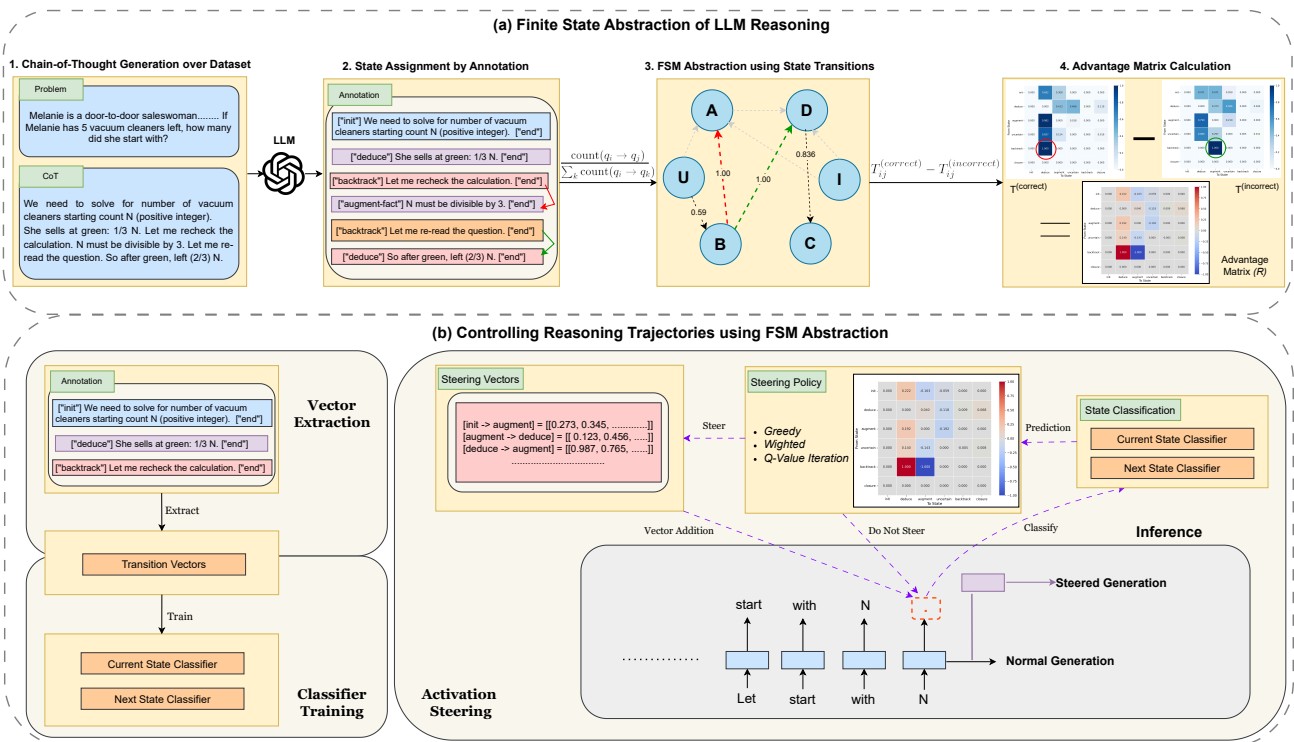

*Figure 2.* Overview of the Finite State Machine (FSM) Abstraction and Steering Framework. **(a)** FSM Abstraction: we first generate Chain-of-Thought (CoT) over a dataset and annotate reasoning steps (e.g., "deduce", "backtrack") to assign states. These annotations are used to construct a probabilistic FSM representing state transitions. We then compute an Advantage Matrix ($R$) by subtracting the transition matrix of incorrect solutions from that of correct solutions. **(b)** Controlling Reasoning Trajectories: to steer generation, we first extract transition vectors and train classifiers to predict the current and next reasoning states. During inference, we classify states at sentence-boundary and a steering policy utilizes the Advantage Matrix to determine optimal transitions and applies activation steering via vector addition if needed to guide the model's reasoning trajectory toward high-value states.

outcome-aligned transition advantage matrix $R \in \mathbb{R}^{|\mathcal{Q}| \times |\mathcal{Q}|}$. Each entry $R_{ij}$ represents the differential likelihood of observing a transition in a correct trajectory versus an incorrect one:

$$R_{ij} = T_{ij}^{(correct)} - T_{ij}^{(incorrect)}$$

The entries $R_{ij}$ capture the difference in transition probabilities between correct and incorrect reasoning trajectories. If $R_{ij} > 0$, the transition $i \to j$ is more frequent in correct reasoning trajectories than in incorrect ones. We term this a positive-advantage transition. It suggests that this move contributes to the robustness of the reasoning process. If $R_{ij} < 0$, the transition is over-represented in incorrect trajectories. This is a negative-advantage transition, indicating a potential failure mode or a suboptimal cognitive shift. Finally, if $R_{ij} = 0$, then we term this as a zero advantage transition, indicating neutral or equally common in both outcomes. Since this matrix relies on a direct empirical estimation of the structural divergence between success and failure, it provides a static, interpretable map of the "terrain" of reasoning correctness, which we will subsequently use for control.

# 3. Controlling Reasoning Trajectories as an FSM Application

Since abstract reasoning states form an outcome-aligned FSM, we use this structure for trajectory control by guiding the model's traversal through the FSM to better match the desirable topology encoded in $T^{(correct)}$. We leverage the transition advantage matrix $R$ as a *control policy prior*: given state $q_t$, the row $R(q_t, \cdot)$ indicates which next-state transitions are outcome-aligned. To steer the model toward a target transition $q \to q'$, following (Chen et al., 2025a), we apply **activation steering** in two steps (Figure 2 (b)): (1) extract a transition steering vector using an off-the-shelf method, and (2) inject it into the hidden representation at sentence boundaries during inference to guide the CoT generation toward the desired state. This provides negligible token overhead and direct control over transition dynamics.

## 3.1. Transition Steering Vectors Extraction

To steer transitions rather than static states, we extract a steering direction for each state pair from the model's hidden activations using the contrastive difference-of-means

method (Turner et al., 2023; Arditi et al., 2024; Venhoff et al., 2025).

**Transition Collection.** For each task, we construct a hold-out set from a separate training split when available, or from the test set otherwise. For each sample, we generate full CoT trajectories using a target model, then segment them into sentence-level units and map them to six predefined states using the automated annotation tool (Section 2). To ensure well-defined transition dynamics, we remove self-loops by merging consecutive identical states into a single state.

**Transition Vector.** For each state $q_k$, we extract the last-token hidden representation at layer $\ell$, denoted $\mathbf{h}_k^{(\ell)} \in \mathbb{R}^d$. Rather than encoding the static state $q_k$, we treat $\mathbf{h}_k^{(\ell)}$ as a **transition vector** representing the directed move from $q_k$ to the next state $q_{k+1}$. Due to the autoregressive nature of LRMs, the last token attends to all prior context, making sentence boundaries the point at which the model effectively commits to the next-state transition.

**Steering Vector.** Given labeled transitions, we define the positive set for a specific transition $t = (u \to v)$ as the set of hidden representations corresponding to that transition across all ordered state pairs with $u \neq v$:

$$\mathcal{D}_{u \to v}^+ := \left\{ \mathbf{h}_k^{(\ell)} \mid (u_k \to v_k) = (u \to v) \right\}$$

Here, k indexes the sentence boundary and $u_k$, $v_k$ denote the source and target states at that position. For each target transition $t = (u \to v)$, we construct a negative set as all other transitions (one-vs-rest at the transition level):

$$\mathcal{D}_{u \to v}^- := \bigcup_{t' \neq (u \to v)} \mathcal{D}_{t'}^+$$

We then compute the positive and negative centroids:

$$\boldsymbol{\mu}_{u \to v}^+ := \mathbb{E}_{\mathbf{h} \sim \mathcal{D}_{u \to v}^+}[\mathbf{h}], \qquad \boldsymbol{\mu}_{u \to v}^- := \mathbb{E}_{\mathbf{h} \sim \mathcal{D}_{u \to v}^-}[\mathbf{h}],$$

and define the transition steering vector as the contrastive difference:

$$\mathbf{v}_{u \to v}^{(\ell)} := \boldsymbol{\mu}_{u \to v}^+ - \boldsymbol{\mu}_{u \to v}^-$$

The steering vector $\mathbf{v}_{u \to v}^{(\ell)}$ identifies hidden-state directions associated with the model being likely to take transition $(u \to v)$, relative to other outgoing transitions.

### 3.2. Deciding When to Steer: State Classification

Even if a desired transition can be induced, intervening at every token can be counterproductive–especially for strong models already on a correct trajectory. We therefore decouple *steering* from *control logic* (when and where to apply them).

Let $\mathbf{h}_t^{(\ell)} \in \mathbb{R}^d$ denote the hidden representation of the $t$-th state at layer $\ell$. We train a State Encoder $f_\theta : \mathbb{R}^d \to \mathbb{R}^k$ to project hidden states into a normalized latent space:

$$\mathbf{z}_t = \frac{f_\theta(\mathbf{h}_t^{(\ell)})}{\|f_\theta(\mathbf{h}_t^{(\ell)})\|}$$

The encoder is trained using triplet loss (Schroff et al., 2015) with margin $m$:

$$\mathcal{L}_{\text{triplet}} = \max\left(0, \|\mathbf{z}_a - \mathbf{z}_p\|_2 - \|\mathbf{z}_a - \mathbf{z}_n\|_2 + m\right),$$

where $(a, p, n)$ are anchor, positive, and negative examples from the same vs. different states, encouraging clustering of same-state embeddings while maintaining inter-state separation. On top of $\mathbf{z}_t$, we train two lightweight classifiers: a **current-state classifier** ($g_{\text{curr}}$) to predict $q_t$, and a **next-state classifier** ($g_{\text{next}}$) to predict $q_{t+1}$ prior to generation.

These classifiers are used only to predict upcoming transitions and compute confidence scores for deciding *whether* to intervene. While state prediction is performed in a learned latent space, steering itself is executed in the original $d$-dimensional hidden representation using $\mathbf{v}_{q \to q'}^{(\ell)}$.

During decoding, classification and steering are applied only at sentence boundaries, detected by punctuation tokens (".", "?", "!"). After each token is generated, we check for punctuation and invoke both classifiers to determine whether intervention is needed. This constraint aligns with the construction of transition steering vectors, which are derived from last-token embeddings, ensuring that control operates at consistent semantic and temporal resolution. Sentence boundaries are well-motivated intervention points: Bogdan et al. (2025) showed sentence-level analysis best captures distinct reasoning steps, while Chauhan et al. (2026) demonstrated punctuation tokens act as semantic summarization boundaries in LLMs. The inability to distinguish sentence-ending punctuation from decimals/equation symbols during autoregressive generation is an inherent constraint of inference-time methods operating at sentence granularity, not specific to our approach.

### 3.3. Identifying Where to Steer: Steering Policy

We interpret the advantage matrix $R$ as a transition-level reward (positive for correctness-aligned transitions, negative for error-prone ones). During decoding, at each sentence boundary punctuation token, given the predicted current state $q$ from $g_{\text{curr}}$, the row $R(q, \cdot)$, and the predicted next state from $g_{\text{next}}$, we select a target $q^\star$ under a chosen policy and apply the corresponding steering vector $\mathbf{v}_{q \to q^\star}^{(\ell)}$.

**(a) Greedy Selection:** This policy always selects the next state with the highest advantage under $R$: $q^* = \arg\max_{q'} R(q, q')$. If $R(q, q^*) > 0$, we steer toward $q^*$.

This strategy follows a locally optimal, myopic decision rule. During decoding, we track the last $L$ states (with $L = 5$); if they are identical, we mark the trajectory as *stuck*, to capture degenerate repetition of the same reasoning mode. If the model is not stuck or the predicted next state already matches $q^*$, we skip steering. While greedy steering exploits $R$ directly, it ignores longer-term transition effects beyond one step.

**(b) $Q$-Value Iteration:** While $R$ captures immediate transition advantages, it ignores long-horizon effects (e.g., transitions that yield moderate short-term gains but lead to highly beneficial future sequences). We therefore treat the FSM as a small planning problem and compute long-horizon transition utilities using a dynamic programming algorithm called $Q$-Value Iteration (Bellman, 1954), considering next-state selection as an action. We iteratively estimate $Q(q, q')$ over 100 steps with discount $\gamma = 0.9$:

$$Q_{k+1}(q, q') := R(q, q') + \gamma \max_{q'' \in Q} Q_k(q', q'')$$

We provide the derivation in Appendix B. Since $R$ is empirically estimated and may contain outliers, we stabilize planning by reward clipping: $R_{\text{clip}}(q, q') := \text{clip}(R(q, q'), [-c, +c]), \quad c \in [0.2, 0.3]$. During inference, given current state $q$, the next-state classifier predicts $\hat{q}_{t+1}$ with probability vector $\mathbf{p}$ and confidence $conf = \max_j p_j$. Using the $Q$-table, we compute: $q^{\star} = \arg\max_{q' \in Q} Q(q, q')$, and $Q_{gap} = Q(q, q^{\star}) - Q(q, \hat{q}_{t+1})$. The $Q$-gap measures how much better the optimal transition is than the model's predicted one. We apply confidence-aware gating with stuck detection: if the model is not stuck and $conf \geq 0.90$, we do not intervene. Otherwise, we steer only when $Q_{\text{gap}} \geq \delta$, directing the model toward $q^{\star}$. Steering strength is adapted dynamically as: $\alpha := \max(\beta, Q_{\text{gap}} \cdot conf), \quad \beta \in [0.1, 1.2]$. This value-propagation policy leverages the FSM abstraction for long-horizon planning, intervening only when the model is likely to deviate from a high-value trajectory. While not guaranteed to be globally optimal (since $Q$ is derived from observed data, not an absolute ground truth), it promotes more coherent long-term reasoning.

**(c) Weighted Steering:** The advantage matrix row $R(q, \cdot)$ often contains multiple informative signals, with some transitions favoring success (positive $R$) and others indicating failure (negative $R$). Instead of selecting a single target transition, we construct a blended steering vector that softly biases the model toward all advantageous transitions, reducing sensitivity to noise in $R$ and avoiding over-commitment to a single path. Given a current state $q$, we define the candidate set $\mathcal{C}(q) = \{q' \in Q : |R(q, q')| \geq \tau\}$ where $\tau = 0$, assign each $q' \in \mathcal{C}(q)$ a signed weight $w(q')$ based on $R(q, q')$, and combine precomputed steering vectors $\mathbf{v}_{q \to q'}^{(\ell)}$

into a single blended steering direction:

$$\mathbf{v}_{\text{weighted}}^{(\ell)}(q) = \sum_{q' \in \mathcal{C}(q)} w(q') \mathbf{v}_{q \to q'}^{(\ell)}.$$

Since weights can be negative, this policy simultaneously **attracts** toward success-aligned transitions and **repels** away from failure-associated ones. We apply confidence gating: if the model is confident and already moving toward a positive $R$ state, we skip intervention; otherwise, we steer.

### 3.4. How to Steer: Orthogonal Transition Injection

Let $\mathbf{h}_k^{(\ell)} \in \mathbb{R}^d$ denote the hidden activation at layer $\ell$ for the last token at a sentence boundary, and let $\mathbf{v}_{u \to v}^{(\ell)}$ be the offline-extracted transition steering direction. Assuming an intervention is required, a naïve approach would be to add $\alpha \mathbf{v}_{u \to v}^{(\ell)}$ directly; instead, we inject only the component of $\mathbf{v}$ orthogonal to the current hidden state to preserve content-aligned components of the hidden state while adding a small perpendicular perturbation that biases next-state generation toward the desired transition. We first normalize: $\hat{\mathbf{h}} = \frac{\mathbf{h}}{\|\mathbf{h}\|_2 + \varepsilon}$, remove the projection of $\mathbf{v}$ onto $\hat{\mathbf{h}}$: $\mathbf{v}_\perp = \mathbf{v} - (\mathbf{v}^\top \hat{\mathbf{h}}) \hat{\mathbf{h}}$, and inject the scaled orthogonal component: $\tilde{\mathbf{h}}_k^{(\ell)} = \mathbf{h}_k^{(\ell)} + \alpha \mathbf{v}_\perp$

## 4. Experimental Setup

**Models.** We evaluate three state-of-the-art open reasoning models: *Qwen3-4B-Thinking* (Yang et al., 2025), *Phi-4-reasoning* (Abdin et al., 2025), and *gpt-oss-20b* (Agarwal et al., 2025). For *gpt-oss-20b*, we run experiments under two reasoning-effort settings (*low* and *medium*) to study how cognitive budget affects transition dynamics. Unless stated otherwise, we refer to these as GPT-L (low), GPT-M (medium), PHI, and QWEN in the remainder of the paper.

**Datasets.** We evaluate reasoning dynamics on four benchmarks spanning domain, difficulty, and reasoning complexity: (1) **AIME25** (MAA Committee, 2025), with 30 challenging math problems; (2) **GPQA Diamond** (Rein et al., 2023), containing 198 expert-validated multiple-choice questions across physics, chemistry, and biology; (3) **MATH-500** (Hendrycks et al., 2021), a 500 math problem test set curated by OpenAI; and (4) **GSM8K** (Cobbe et al., 2021), a grade-school math word problem dataset with 1319 test problems.

**Evaluation Metrics.** Our primary metric is *accuracy* (percentage of correctly solved problems). We also report *average CoT length in tokens* to assess changes in verbosity and the *average number of steering interventions* per CoT to measure intervention frequency. Steering is applied only during CoT generation, tracked via each model's thinking start and end tokens.

*Table 1.* Comparison of model performance under different steering strategies across four benchmarks. Metrics include Accuracy (%), average CoT generation length (Avg #Tokens), and average steering frequency (Avg #Steering), highlighting trade-offs between correctness, verbosity, and intervention cost. The best score for accuracy is shown in **bold**, and the second-best is underlined.

| Dataset | Model | Default | | Greedy Steering | | | Weighted Steering | | | $Q$-Value Steering | | |
|---|---|---|---|---|---|---|---|---|---|---|---|---|
| | | Accuracy | Avg #Tokens | Accuracy ↑ | Avg #Tokens ↓ | Avg #Steering ↓ | Accuracy ↑ | Avg #Tokens ↓ | Avg #Steering ↓ | Accuracy ↑ | Avg #Tokens ↓ | Avg #Steering ↓ |
| AIME 25 | GPT-L | 43.30 | 1725.13 | 50.00 | 1598.90 | 77.60 | 56.67 | 1852.43 | 83.87 | **56.67** | 2200.27 | 55.20 |
| | GPT-M | 66.67 | 10507.70 | 66.67 | 9823.73 | 498.13 | 76.67 | 10054.47 | 501.77 | **76.67** | 11201.73 | 270.73 |
| | PHI | 73.33 | 10660.23 | 73.33 | 10460.37 | 328.13 | 76.67 | 9739.87 | 171.20 | **76.67** | 9710.83 | 75.50 |
| | QWEN | 83.33 | 19795.93 | 76.67 | 19426.03 | 287.13 | 76.67 | 19954.87 | 151.53 | **86.67** | 19747.33 | 42.40 |
| MATH 500 | GPT-L | 79.00 | 286.81 | 81.20 | 282.34 | 12.17 | 82.40 | 298.10 | 12.81 | **83.20** | 279.75 | 0.48 |
| | GPT-M | 86.40 | 1202.69 | 85.40 | 945.95 | 42.69 | 86.20 | 1166.51 | 54.65 | **87.00** | 1091.57 | 0.30 |
| | PHI | 89.00 | 1391.16 | 88.20 | 1346.01 | 34.05 | **90.80** | 1567.03 | 45.53 | 89.60 | 1541.07 | 9.74 |
| | QWEN | 91.00 | 4152.48 | 91.00 | 3809.25 | 76.10 | 91.00 | 4181.20 | 47.23 | **91.40** | 4182.82 | 16.78 |
| GPQA Diamond | GPT-L | 57.57 | 332.50 | **63.64** | 354.40 | 16.27 | 60.10 | 389.67 | 24.55 | 62.12 | 362.40 | 7.42 |
| | GPT-M | 64.14 | 4111.91 | 64.14 | 5177.40 | 246.93 | 65.15 | 4676.32 | 183.74 | **67.17** | 4962.95 | 88.12 |
| | PHI | 67.17 | 7719.51 | 67.17 | 3672.36 | 145.30 | **71.72** | 7359.44 | 184.64 | 69.19 | 7367.14 | 38.18 |
| | QWEN | 62.12 | 7770.20 | 62.63 | 7770.20 | 206.30 | **65.15** | 7770.20 | 146.31 | 63.13 | 7770.20 | 58.32 |
| GSM8K | GPT-L | 94.39 | 84.05 | 93.74 | 72.45 | 4.87 | 94.62 | 73.67 | 3.00 | **94.62** | 73.34 | 2.38 |
| | GPT-M | 95.60 | 297.34 | 95.07 | 247.40 | 15.93 | 95.68 | 249.12 | 12.41 | **95.91** | 250.00 | 4.28 |
| | PHI | 96.44 | 615.67 | 96.06 | 545.27 | 19.30 | 96.44 | 541.97 | 11.94 | **96.51** | 538.20 | 5.57 |
| | QWEN | 78.77 | 1067.53 | 77.48 | 1067.53 | 40.39 | 79.23 | 1067.53 | 15.17 | **79.30** | 1067.53 | 6.05 |

**Annotation.** Following (Venhoff et al., 2025), we employ *GPT-4o-mini* with a near-deterministic temperature of $1 \times 10^{-19}$. Annotation prompts and examples are provided in Appendix K and Appendix M. To validate labeling quality, we manually reviewed 10% of AIME25 and GPQA Diamond annotations with two independent raters, achieving high inter-annotator agreement (Cohen's Kappa = 0.89 (Cohen, 1960)), indicating strong consistency between automated and human labels.

**Hyper-parameter Settings.** During inference, we use each model's default generation sampling parameters and set the maximum output length to 32,768 tokens. All experiments are implemented in PyTorch and run with a fixed random seed of 42 for reproducibility. We train the state encoder with a 512-dimensional projection head using a two-layer MLP (LayerNorm, ReLU, dropout = 0.1) and $\ell_2$ normalization to embed representations on a unit hypersphere, optimized with triplet margin loss (margin = 1.1) using Adam (lr = $10^{-4}$) for 50 epochs. Using these embeddings, we train separate MLP classifiers for current- and next-state prediction with an 80–20 train–test split, achieving over 90% test accuracy across all settings. Steering vectors are $\ell_2$-normalized with $\epsilon = 10^{-8}$. Following (Chen et al., 2025b), we extract steering vectors from multiple layers and select the best-performing layer on a hold-out set: GPT-L/GPT-M at layer 19, PHI at layer 22, and QWEN at layer 30. We set $\alpha = 1.0$ for Greedy steering, $\alpha \in [0.1, 1.0]$ for Weighted steering, and a $Q$-gap threshold of $\delta = 0.06$.

## 5. Experimental Results

We evaluate FSM-based reasoning control by comparing Greedy, Weighted, and $Q$-Value steering against a no-steering (default) baseline. As shown in Table 1, all of our steering methods improve accuracy, though methods differ in effectiveness and intervention cost. Figure 3 provides qualitative examples showing that steering can redirect the reasoning trajectory at critical moments (e.g., where deduction is more important than knowledge augmentation) to reach correct solutions where the default behavior fails.

**Performance Gains in Reasoning Tasks.** Steering yields the maximum gains on high-difficulty tasks such as AIME25. For GPT-L, accuracy rises from 43.3% to 56.67% under both weighted and $Q$-Value steering, but $Q$-Value steering achieves this with substantially fewer interventions (55.20 vs. 83.87 on average). A similar trend is observed on MATH-500, where GPT-L reaches a peak accuracy of 83.20%, outperforming greedy steering (81.20%), suggesting that $Q$-Value steering intervenes only at critical decision points. Consistent improvements also appear in scientific and logical reasoning: on GPQA Diamond, GPT-M improves from 64.14% to 67.17% with $Q$-Value steering. Even on simpler tasks like GSM8K, where default baseline accuracy is already high, $Q$-Value steering maintains or slightly improves performance (e.g., QWEN reaching 79.30%).

**Token Efficiency and CoT Length.** Beyond accuracy, FSM-based steering influences *how* models reason. We observe that $Q$-Value steering often achieves equal or higher accuracy with comparable or fewer tokens, indicating more efficient reasoning trajectories. On AIME25, weighted steering frequently increases token usage (e.g., GPT-L: 1852 vs. 1725 default), whereas $Q$-Value steering may sometime generate longer CoTs but with substantially fewer interventions. For QWEN, $Q$-Value steering achieves higher accuracy (86.67%) with token counts similar to the default baseline. On MATH-500 and GSM8K, $Q$-Value steering often reduces average CoT tokens relative to default and greedy steering (e.g., GPT-L on GSM8K: 84.05 to 73.34), sug-

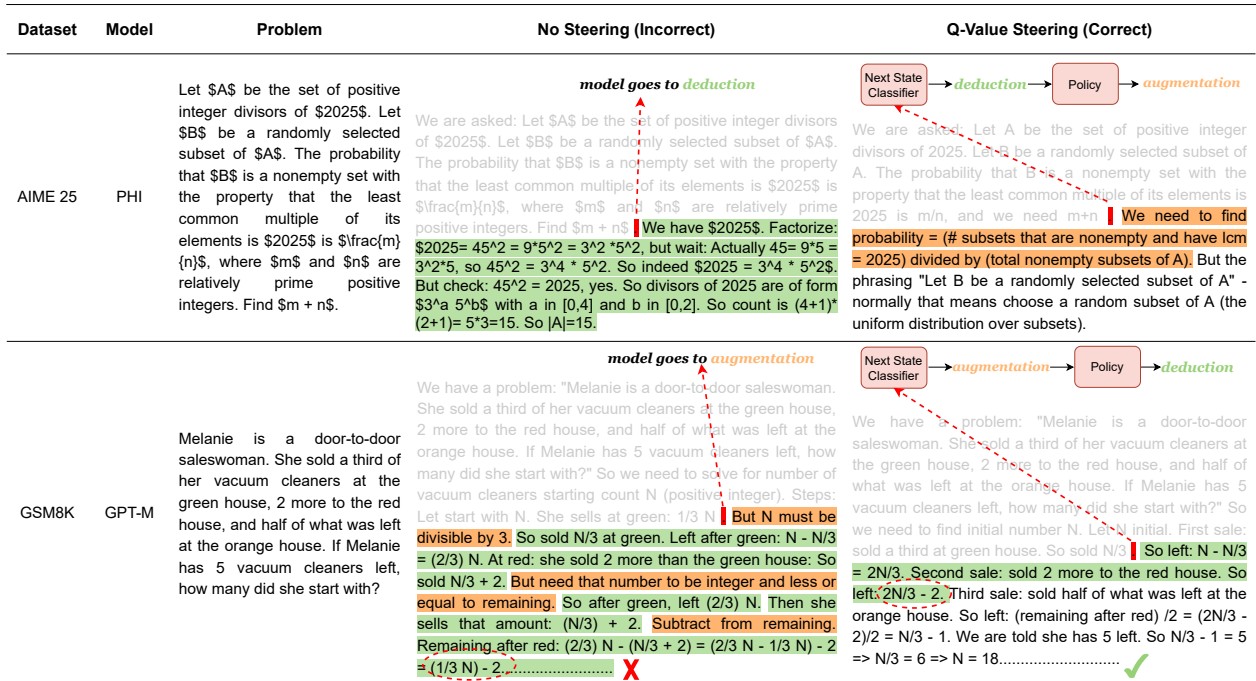

*Figure 3.* Qualitative examples of how steering corrects erroneous reasoning trajectories during problem solving. In the bottom example, without steering, the model follows a flawed transition sequence and produces incorrect intermediate steps. However, with steering-based control, the next-state classifier detects suboptimal transitions (e.g., model is preparing to do "augmentation") and triggers policy-driven intervention, redirecting the model toward a more reliable reasoning state (e.g., continue "deduction"), yielding correct intermediate answers.

gesting faster convergence to correct solutions. On GPQA Diamond, token counts remain relatively stable across methods due to task complexity, but $Q$-Value steering avoids the extreme token inflation observed in some greedy configurations (e.g., GPT-M).

**Intervention Efficiency.** Our analysis reveals that steering frequency directly affects task accuracy. Across nearly all models and datasets, $Q$-Value steering matches or outperforms other methods while requiring the fewest interventions. On MATH-500, GPT-L reaches peak performance with only 0.476 interventions per sample, compared to 12.17 under greedy setting (a ∼25× reduction), indicating that value-based policy identifies high-leverage decision points in a reasoning trajectory where a correction is most impactful, rather than biasing the model's distribution at every token. Compared to weighted steering–which often incurs more steering steps and longer CoTs–$Q$-Value steering maintains competitive CoT lengths (e.g., on AIME25) while significantly reducing computational overhead.

**Comparative Analysis of Steering Methods.** Our findings reveal that short-sighted steering strategies can degrade performance. For instance, greedy steering, while simple, sometimes reduces accuracy (e.g., QWEN on AIME25: 83.3% → 76.67%; GPT-M on MATH-500: 86.40% → 85.40%), indicating that pushing the model toward local

high-reward states can lead the reasoning process into suboptimal paths that are difficult to correct later in long reasoning trajectory. In contrast, $Q$-Value steering provides a more robust alternative by leveraging cumulative future-reward estimates to avoid over-steering.

In addition to our three activation-based steering methods, we evaluate a prompt-based steering approach that leverages the advantage matrix to explicitly guide the model's reasoning by specifying desired and undesired behaviors. While activation-based steering consistently outperforms prompt-based steering, the latter still achieves meaningful performance gains, indicating that it is a promising complementary direction for future work. For brevity, we defer details to the Appendix D.

**Cross Model Transferability.** Cross-model transferability is partially addressed in Table 5 of Appendix D via prompt-based Cross-Model Guidance. We additionally evaluate cross-model activation steering by applying QWEN's advantage matrix to GPT-L over the MATH-500 dataset, comparing performance against model-specific steering methods. Table 2 shows that cross-model steering remains competitive across all three conditions. Under greedy steering, cross-model steering marginally outperforms the model-specific baseline (81.80% vs. 81.20%), suggesting that QWEN's transition dynamics encode reasoning patterns that transfer

*Table 2.* Cross-model transferability results on GPT-L using QWEN's advantage matrix over MATH-500 dataset. Metrics follow the same notation as Table 1. The best accuracy score is shown in **bold**.

| Setup | Greedy | | | Weighted | | | Q-Value | | |
|---|---|---|---|---|---|---|---|---|---|
| | Accuracy ↑ | Avg #Tokens ↓ | Avg #Steering ↓ | Accuracy ↑ | Avg #Tokens ↓ | Avg #Steering ↓ | Accuracy ↑ | Avg #Tokens ↓ | Avg #Steering ↓ |
| Model Specific | 81.20 | 282.34 | 12.17 | **82.40** | 298.10 | 12.81 | **83.20** | 279.75 | 0.48 |
| Cross Model | **81.80** | 307.18 | 17.06 | 81.60 | 296.00 | 14.58 | 82.80 | 293.85 | 5.37 |

non-trivially to GPT-L. The results on MATH-500 dataset suggest that cross-model transition dynamics are *partially universal*: a transferred advantage matrix preserves enough signal to meaningfully improve over the unsteered baseline, but loses the fine-grained calibration needed to match model-specific steering performance. The accuracy degradation is modest, while the intervention cost is substantially higher.

# 6. Related Work

**Modeling and Controlling CoT as Cognitive Dynamics.** Early Large Language Models (LLMs) struggled on complex reasoning benchmarks (Mirzadeh et al., 2025; Nezhurina et al., 2024), motivating Chain-of-Thought (CoT) prompting (Wei et al., 2022; Nye et al., 2021; Reynolds & McDonell, 2021) and self-verification (Weng et al., 2023; Zhao et al., 2025) to elicit intermediate reasoning steps. Recent work has emphasized *structuring* and *interpreting* CoT reasoning: Li et al. (2025) assigns cognitive labels to CoT, Besta et al. (2024) models reasoning as graphs of thoughts, and Lee et al. (2025) parses trajectories into DAG-based reasoning flows.

A complementary line of work analyzes reasoning structure through graph-theoretic properties of hidden-state trajectories: Minegishi et al. (2026) extracts reasoning graphs by clustering LLM hidden states and finds that distilled reasoning models exhibit significantly more cyclic structures, larger diameters, and stronger small-world properties than base models, with these structural advantages correlating with accuracy; Matsutani et al. (2025) further shows that RL compresses incorrect trajectories while SFT expands correct ones, and that RL concentrates graph functionality into hub nodes while SFT distributes it broadly. At a more semantic level, Xiong et al. (2025) constructs reasoning graphs from raw CoT tokens via LLM-guided clustering and shows that graph-level metrics such as exploration density and branching ratio strongly predict task success, and that few-shot prompting degrades performance precisely by flattening these structural properties.

CoT is increasingly viewed as a trajectory through reusable reasoning behaviors that can be *measured* and *controlled*. For example, Chen et al. (2025a) decomposes CoT into execution, reflection, and transition modes, showing that

suppressing unproductive categories via training-free latent steering improves accuracy and efficiency; however, their control remains category-level (always suppressing reflection/transition) without explicitly deciding *which* cognitive transition should occur next under long-horizon objectives. Similarly, Venhoff et al. (2025) shows that certain reasoning behaviors are mediated by linear directions in the activation space and can be causally modulated at inference time, but intervenes on isolated behaviors without modeling *which state transitions* best support end-task correctness under a global objective.

**Our Contribution: Planning-Aware Control over CoT Trajectories.** While prior work offers mechanisms for local behavioral adjustment, a fundamental gap remains: deciding *when* and *where* to intervene along long trajectories to maximize task success. We bridge this gap by formalizing CoT as a Finite State Machine (FSM) over six high-level cognitive states and modeling reasoning as traversal dynamics in this discrete state space. Rather than steering a single behavior in isolation, we define steering policies over the FSM transition graph to derive an optimal control policy that identifies which cognitive shifts (e.g., moving from uncertainty to backtracking) are statistically aligned with correct outcomes, enabling sparse and strategic interventions via orthogonal activation steering applied only at critical transition boundaries.

# 7. Conclusion

In this work, we demonstrate that the complex reasoning of Large Language Models can be effectively modeled and controlled using a simple Finite State Machine. By mapping Chain-of-Thoughts into just six distinct thinking stages–such as deduction or backtracking–we uncovered clear structural differences between successful and failed reasoning strategies. We leveraged this insight to create a steering mechanism that acts like a GPS for the model, using long-term planning (*Q*-Value Steering) to nudge it toward the most promising next step only when necessary. Our experiments show that this approach significantly boosts performance on difficult tasks with very few interventions, proving that we can make LRMs more robust by guiding the high-level strategy behind the thoughts.

## Impact Statement

This work advances the interpretability and controllability of Large Reasoning Models (LRMs). By formalizing reasoning as a Finite State Machine, we provide a mechanism to not only observe but actively guide the cognitive processes of AI systems. The primary positive impact of this research is the potential for enhanced reliability and safety in high-stakes domains. As LRMs are increasingly deployed in fields like software engineering, scientific research, and financial modeling, the ability to detect degenerate reasoning steps and steer models away from logical fallacies is critical for preventing costly or dangerous errors. Furthermore, the FSM abstraction offers a layer of transparency, allowing users to audit the "strategy" a model employed—verifying not just the final answer, but the structural validity of the path taken to reach it.

However, the capability to steer reasoning trajectories also introduces risks regarding cognitive rigidity and bias. Our steering policies are derived from CoTs generated for existing benchmarks which are then annotated by specific frontier models (e.g., GPT-4o-mini). Aggressively steering towards these outcome-aligned patterns could inadvertently suppress diverse or unconventional problem-solving approaches that do not fit the predefined state topology.

Finally, while our focus is on mathematical and logical reasoning, the general capability to enhance chain-of-thought robustness contributes to the broader trend of increasing AI autonomy. As models become better at long-horizon planning and self-correction, necessary safeguards must be developed to ensure these capabilities are not exploited for malicious purposes, such as automated vulnerability exploitation or disinformation generation. We believe our interpretability framework serves as a step toward such safeguards, providing the monitoring tools necessary to detect and interrupt anomalous reasoning behaviors.

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

# Appendix

## A. Description of FSM States

**(1) Initialization (init):** The model is interpreting the question or setting up the problem. In this state, the text span is typically a restating or rephrasing of the task, clarification of what is being asked, or a statement of approach (i.e. "Okay, we are given..."). Formally, we can think of *init* as the starting state for the reasoning FSM in most cases as it captures any preliminary thinking before actual solving steps begin. However, not all CoTs explicitly have an initialization step; sometimes a model dives straight into solving (*deduce* or *augment*) without rephrasing the problem. Our framework allows any state (except *closure*) to technically be a start.

**(2) Deduction (deduce):** The model is performing logical steps based on known information. In this state, the text span often includes inferential steps, calculations, drawing implications or making intermediate conclusions from premises. For example, lines of mathematical work, factual inferences ("$X$ implies $Y$"), or any straightforward reasoning step belong here. We can consider *deduce* as the "default" problem-solving state.

**(3) Augmentation Strategy (augment):** The model is using an auxiliary strategy to strengthen or extend its reasoning. We further divide this category into the following subtypes:

(a) **Adding Knowledge (augment-fact):** Injecting external knowledge or recalling a fact or internal knowledge not explicitly given in the prompt (i.e. "I know that Paris is the capital of France, which might be relevant here.").

(b) **Example Testing (augment-test):** Trying a specific example or test case to gain insight or verify (e.g., "Let's test this with a simple number: if $x = 2$, then ...").

(c) **Explore Alternative Paths (augment-branch):** Exploring a different line of reasoning than initially pursued (i.e. "Alternatively, perhaps we should consider another scenario where ...").

(d) **Planning Solution (augment-plan):** Laying out a plan or high-level outline of the solution approach (i.e. "First, I will do X. Then, I will check Y. After that, Z.").

(e) **Refinement Strategy (augment-refine):** Engaging in self-reflection or self-correction or self-verification, merging previous steps, double checking calculations, summarizing key points, preparing final answer, or re-confirming its own reasoning (i.e. "let me rewrite the definition in a simpler way" or "Let me double-check the earlier calculation.").

(f) **Emergent Strategy (augment-emerge):** Any other strategy that does not fit the above, possibly an unforeseen creative tactic the model employs.

All these sub-states are encompassed by *augment* state. They represent moments where the model steps outside of straightforward deduction and employs a deliberate tactic to explore. In our FSM framework, we could depict *augment* as a composite state with internal distinctions, but for simplicity, we treat it as one state with an attribute indicating which subtype (a–f) is used.

**(4) Uncertainty Estimation (uncertain):** The model explicitly expresses doubt, confusion, hesitation or lack of confidence about its current step, assumptions, or calculations. The model may acknowledge missing information, ambiguous problem framing, review again or the possibility that the deduction could be wrong. The text spans in this state might include phrases such as "I'm not sure", "This is confusing", or "Maybe I did a mistake somewhere".

**(5) Backtracking (backtrack):** The model goes back to a previous step/assumption, or re-reads the question or instructions, or re-evaluates earlier calculations often following an uncertainty or realization of error. In this state, the text span could be a repetition or paraphrase of part of the prompt, or a statement like "Let me re-read the problem", "Let's calculate the 3rd step again" or "Going back to what was stated initially...". Entering this state implies the model is not currently progressing forward with new deductions, but rather reviewing prior assumptions or calculations.

**(6) Final Conclusion (closure):** The model decides a final answer or option to the query or output action, either directly or as a meta-statement. This is a terminating (accepting) state in automata terms. Once in *closure* state, the reasoning ends and the LRM starts to generate final answer based on the CoT. The text span here is usually a concise answer or summary of the solution, possibly with a phrase such as "Therefore, the answer is ..." or "So, Option: C", or "Okay, I am ready to write the final answer". By definition, each complete reasoning trajectory should have *closure* at the very end.

## B. Value-based Policy: Deterministic MDP Reduction

Standard optimal $Q$-iteration in a Markov Decision Process (MDP) is defined over $(Q, A, T, R, \gamma)$ with:

- $Q$: states
- $A$: actions
- $T(q, a, q') = P(q' \mid q, a)$: transition probabilities
- $R(q, a, q')$: reward
- $\gamma$: discount factor

The optimal Bellman equation is defined as:

$$Q^*(q, a) = \sum_{q'} T(q, a, q') \Big( R(q, a, q') + \gamma \max_{a'} Q^*(q', a') \Big) \qquad (1)$$

Our case is a deliberate reduction where planning happens over the abstract reasoning FSM:

- **States ($Q$):** The 6 reasoning states (e.g., *init, augment, deduce, uncertain, backtrack, closure*).

- **Action Space ($A = Q$):** Choosing action $a = q'$ means steer toward next state $q'$.

- **Deterministic Steering:** Steering selects the next abstract state, so:

$$T(q, a = q', q') = 1 \quad \text{and} \quad T(q, a = q', \tilde{q} \neq q') = 0 \qquad (2)$$

- **Reward ($R_{\text{clip}}$):** The immediate reward for taking action $a = q'$ in state $q$ is the transition advantage. We use a clipped version for stability:

$$R_{\text{clip}}(q, a = q') \approx R(q, q') \quad \text{(clipped)} \qquad (3)$$

- **Discount Factor ($\gamma$):** Controls the lookahead depth.

Plugging the deterministic $T$ into the standard equation (1) collapses the sum over $q'$ (because only one next state has probability 1), yielding:

$$Q^*(q, a = q') = R_{\text{clip}}(q, q') + \gamma \max_{a'} Q^*(q', a') \qquad (4)$$

Since $A = Q$, we can rewrite $\max_{a'}$ as a maximum over next states $q'' \in Q$:

$$Q^*(q, q') = R_{\text{clip}}(q, q') + \gamma \max_{q'' \in Q} Q^*(q', q'') \qquad (5)$$

Here, we can interpret $R_{\text{clip}}(q, q')$ as "how correlated this transition is with correctness right now", $\max_{q''} Q^*(q', q'')$ as "best achievable future correctness after landing in $q'$", and $Q^*(q, q')$ as "this step's outcome-alignment + best future outcome-alignment afterward". This is exactly why planning adds value: it prefers transitions that lead into "good regions" of the reasoning dynamics, not just those with high immediate reward in $R$.

## C. Empirical Validation of Markov Assumption

We acknowledge that the Markov assumption is a first-order approximation because it does not capture the state of the model. We considered memory-based models such as push-down automata but discounted them because the context of a model is significantly different from a stack. We hope to explore more sophisticated models in our future work.

To empirically validate the impact of the Markovian assumption, we conducted a second-order analysis on MATH-500 dataset. To analyze, we experimented using all four models across all three steering methods. For Greedy and Weighted intervention, the steering policy uses second-order advantage matrix $R^2[(\text{prev}, \text{curr}, \text{next})]$ with first-order matrix $R^1$ as fallback for structurally absent pairs. For Q-Value steering, we tested three variants:

1. **Fallback:** immediate reward from $R^2$ with future value from $R^1$-based Q-table,

2. **Interpolation:** blended $\lambda \cdot R^2 + (1 - \lambda) \cdot R^1$ where $\lambda = \min(\text{count}/\text{threshold}, 1)$. We consider count = number of distinct prev states for which $R^2[(\text{prev}, \text{curr}, \text{next})]$ is observed and threshold = 4,

3. **Full Second-Order:** immediate reward from $R^2$ with $R^2$-mean (mean of $R^2$ over observed prev states, $R^1$ fallback for absent pairs) Q-table.

In all variants, previous state is tracked at inference with no new classifier needed.

*Table 3.* Results of Greedy and Weighted second-order steering. Acc = Accuracy (%), AT = Average number of tokens, AS = Average number of steering actions, GFO = Accuracy gain of second-order over first-order.

| | Greedy | | | | Weighted | | | |
|---|---|---|---|---|---|---|---|---|
| Model | Acc | AT | AS | GFO | Acc | AT | AS | GFO |
| GPT-L | 81.80 | 328.74 | 17.32 | +0.60 | 79.80 | 291.50 | 14.24 | −2.60 |
| GPT-M | 85.40 | 1109.70 | 39.46 | 0.00 | 85.80 | 1251.07 | 40.54 | −0.80 |
| PHI | 90.00 | 1379.74 | 38.87 | +0.80 | 89.40 | 1388.83 | 37.87 | −1.40 |
| QWEN | 91.00 | 4159.25 | 82.49 | 0.00 | 90.60 | 4188.78 | 45.22 | −0.40 |

*Table 4.* Q-Value second-order steering across all three variants: Fallback, Interpolation, and Full Second-Order. Column headers follow the same notation as Table 3.

| | Fallback | | | | Interpolation | | | | Full Second-Order | | | |
|---|---|---|---|---|---|---|---|---|---|---|---|---|
| Model | Acc | AT | AS | GFO | Acc | AT | AS | GFO | Acc | AT | AS | GFO |
| GPT-L | 82.80 | 302.10 | 6.37 | −0.40 | 82.00 | 315.32 | 6.72 | −1.20 | 82.60 | 315.19 | 7.71 | −0.60 |
| GPT-M | 86.00 | 1248.92 | 13.73 | −1.00 | 85.40 | 1191.57 | 14.39 | −1.60 | 85.60 | 1205.04 | 9.94 | −1.40 |
| PHI | 89.40 | 1468.57 | 4.66 | −0.20 | 89.40 | 1466.54 | 4.47 | −0.20 | 88.80 | 1485.73 | 4.02 | −0.80 |
| QWEN | 91.20 | 4226.58 | 17.95 | −0.20 | 91.00 | 4246.81 | 17.86 | −0.40 | 91.40 | 4251.58 | 17.01 | 0.00 |

The empirical comparison across five steering configurations and four models consistently shows that second-order modeling yields little or no accuracy benefit over the first-order Markov design choice. If the second-order information were genuinely capturing additional structure beyond first-order, we would expect consistent positive GFO values. Instead, GFO is slightly positive only for Greedy (GPT-L: +0.60, PHI: +0.80) and largely neutral or slightly negative elsewhere.

To further validate, we conducted a sign agreement analysis on MATH-500 benchmark using *Qwen3-4B-Thinking* by:

1. computing first and second-order transition matrices for correct/incorrect trajectories,

2. deriving $R$ and $R^2$, then comparing per-triplet sign agreement for every valid (prev, curr, next),

3. checking $\text{sign}(R[\text{curr,next}]) == \text{sign}(R^2[\text{prev} \rightarrow \text{curr}, \text{next}])$.

**Sign Agreement: 82.1% (64/78 triplets)** – first-order $R$ correctly identifies beneficial transition direction in 4/5 contexts. The 14 disagreements concentrate on near-zero $R$ values (mean $|R| \approx 0.01$) – neutral transitions where steering is not triggered. **Pearson r=0.63, Spearman r=0.62** (p<1e-9) confirm $R$ and $R^2$ are strongly correlated. **Mean entropy** $\Delta H$=0.019 **bits** – second-order adds negligible predictive information beyond first-order.

We believe that this direction remains promising; for example, we may find different advantage matrices that correspond to different stages of the reasoning (early vs. late) or that are based on the nature of the reasoning task, and we leave a more thorough exploration of this to future work.

# D. Prompt-based Steering

In this section, we investigate whether the performance gains of our FSM framework arise from latent $Q$-value activation steering, or whether similar improvements can be achieved by exposing FSM transition structure directly in the prompt as natural-language guidance. This isolates the effect of semantic instruction from latent intervention. Using the transition advantage matrix $\mathbf{R}$, we construct *four* prompt-only steering variants, where transition preferences are provided as textual constraints rather than numerical biases:

- **Positive Guidance**: The model is instructed to preferentially transition toward the state with the highest positive advantage value (e.g., from "deduce" state, move toward the most beneficial next state).

- **Negative Guidance**: The model is instructed to explicitly avoid transitioning to the state with the most negative advantage value (largest magnitude among negative entries).

- **Balanced Guidance**: A combined strategy that both encourages the highest-advantage transition and discourages the most negative one.

- **Cross-Model Guidance**: To test transferability of reasoning structure, we apply the Balanced Guidance prompt derived from the best-performing model to all other models, evaluating whether FSM-induced reasoning patterns generalize across architectures.

We report Accuracy and the average Chain-of-Thought (CoT) generation length across four benchmarks in Table 5, comparing default inference, $Q$-Value activation steering, and the four prompt-based alternatives. For a detailed breakdown of our steering instructions, we provide the prompt templates used for GPT-M on MATH-500 dataset in Appendix L; configurations for other models and datasets follow a standardized format based on these representative templates.

*Table 5.* Comparison of model performance under different FSM-based steering strategies across four benchmarks. Metrics include Accuracy (%) and average Chain-of-Thought length (Avg #Tokens). We compare default inference, $Q$-Value activation steering, and prompt-based transition guidance (Positive, Negative, Balanced, and Cross-Model). The best accuracy for each model is shown in **bold**, and the second-best is underlined. Dashes (–) indicate models that serve as the source for Cross-Model Guidance rather than a target.

| Dataset | Model | Default | | Q-Value Steering | | Positive Guidance | | Negative Guidance | | Balanced Guidance | | Cross-Model Guidance | |
|---|---|---|---|---|---|---|---|---|---|---|---|---|---|
| | | Accuracy | Avg #Tokens | Accuracy ↑ | Avg #Tokens ↓ | Accuracy ↑ | Avg #Tokens ↓ | Accuracy ↑ | Avg #Tokens ↓ | Accuracy ↑ | Avg #Tokens ↓ | Accuracy ↑ | Avg #Tokens ↓ |
| AIME 25 | GPT-L | 43.30 | 1725.13 | **56.67** | 2200.27 | 36.67 | 481.43 | 33.33 | 318.00 | 26.67 | 298.23 | 46.67 | 637.63 |
| | GPT-M | 66.67 | 10507.70 | **76.67** | 11201.73 | 66.67 | 5383.87 | 70.00 | 4261.17 | 73.33 | 4600.00 | 66.67 | 5435.70 |
| | PHI | 73.33 | 10660.23 | **76.67** | 9710.83 | 73.33 | 7032.20 | 60.00 | 4179.53 | 53.33 | 7028.60 | 53.33 | 4048.93 |
| | QWEN | 83.33 | 19795.93 | **86.67** | 19747.33 | 73.33 | 19782.43 | 76.67 | 12130.00 | 70.00 | 20526.27 | – | – |
| MATH 500 | GPT-L | 79.00 | 286.81 | **83.20** | 279.75 | 80.40 | 181.78 | 78.60 | 162.51 | 81.20 | 168.08 | 80.60 | 183.79 |
| | GPT-M | 86.40 | 1202.69 | **87.00** | 1091.57 | 86.00 | 838.10 | 86.20 | 663.31 | 86.20 | 583.26 | 85.60 | 640.89 |
| | PHI | 89.00 | 1391.16 | **89.60** | 1541.07 | 89.20 | 1487.70 | 89.40 | 1216.48 | 89.40 | 1547.54 | 87.40 | 1308.53 |
| | QWEN | 91.00 | 4152.48 | 91.40 | 4182.82 | 91.20 | 3267.40 | 91.20 | 2511.23 | **91.40** | 2698.12 | – | – |
| GPQA Diamond | GPT-L | 57.57 | 332.50 | 62.12 | 362.40 | 54.55 | 182.29 | 62.12 | 218.19 | **63.64** | 230.84 | 56.06 | 160.01 |
| | GPT-M | 64.14 | 4111.91 | 67.17 | 4962.95 | **69.19** | 2368.94 | 66.67 | 1196.11 | 64.65 | 1287.29 | 66.16 | 1462.09 |
| | PHI | 67.17 | 7719.51 | **69.19** | 7367.14 | 66.16 | 3993.25 | 66.16 | 3487.99 | 64.14 | 3734.92 | – | – |
| | QWEN | 62.12 | 7770.20 | 63.13 | 7770.20 | 65.15 | 4489.04 | 64.65 | 5042.89 | 66.16 | 5138.22 | **66.16** | 4914.22 |
| GSM8K | GPT-L | 94.39 | 84.50 | **94.62** | 73.34 | 94.16 | 72.16 | 93.63 | 64.23 | 93.40 | 68.54 | 93.93 | 66.88 |
| | GPT-M | 95.60 | 297.34 | **95.91** | 250.00 | 95.38 | 245.22 | 95.07 | 232.06 | 95.15 | 197.18 | 95.07 | 239.60 |
| | PHI | 96.44 | 615.67 | **96.51** | 538.20 | 95.30 | 1072.88 | 96.13 | 1300.37 | 96.13 | 1345.00 | – | – |
| | QWEN | 78.77 | 1067.53 | 79.30 | 1067.53 | 93.78 | 1461.96 | **94.77** | 1469.04 | 94.16 | 1464.31 | 94.31 | 1750.67 |

Our analysis reveals several key findings:

**(a) Latent Steering Is More Effective.** Across most settings—especially on AIME25 and MATH 500—*Q-Value steering consistently outperforms prompt-based guidance*. While prompts can describe FSM transition logic, they often fail to enforce it reliably during multi-step reasoning. For example, on AIME 25 with GPT-L, $Q$-Value steering achieves 56.67% accuracy, whereas the best prompt-based method (Cross-Model Guidance) reaches only 46.67%. This suggests that direct latent intervention provides stronger and more dependable control than textual instruction.

**(b) Efficiency Without Sacrificing Accuracy.** We also observe a consistent effect on Chain-of-Thought (CoT) length. Prompt-based methods—especially Balanced and Negative Guidance—often *reduce CoT length substantially*, but this reduction frequently comes at the cost of accuracy. In contrast, $Q$-Value steering maintains or improves correctness while

keeping token usage relatively stable, indicating that it compresses reasoning by increasing information density rather than prematurely shortening the CoT.

**(c) Instruction Overload in Prompt Guidance.** In some cases, prompt-based steering can help (e.g., QWEN on GSM8K improves from 78.77% to 94.77%). However, for many models, adding complex transition rules to the prompt *hurts performance* (e.g., PHI on AIME25 drops from 73.33% to 53.33% under Balanced Guidance). This suggests that high-level semantic constraints can interfere with the model's natural reasoning process, while latent steering avoids this conflict by operating directly within the model's internal dynamics.

**(d) Limited Cross-Model Transfer.** Cross-Model Guidance shows that some reasoning structure can transfer across models, but performance remains below that of model-specific latent steering. This indicates that FSM transition patterns are partially model-dependent, and the most effective control arises when FSM logic is embedded directly into a model's internal decision process rather than imposed externally.

## E. Limitations and Future Work

This section outlines some limitations of our approach and potential directions for future work.

**Coarse, Fixed, and Imperfect State Abstraction.** We model long reasoning as a memoryless FSM over six discrete states. This makes the analysis and control tractable, but it cannot capture finer-grained skills (e.g., algebraic manipulation vs. proof search) or longer-range dependencies where the best next move depends on earlier context, not just the current state. In addition, the six-state taxonomy was chosen for math and science tasks and may not transfer cleanly to other domains (e.g., creative writing, legal reasoning, open-ended coding) where additional or different states may be needed. Finally, state labels come from automated annotation (GPT-4o-mini) and learned classifiers; any annotation bias or classification error can propagate to the transition statistics and trigger unnecessary or harmful interventions.

**From Correlation to Cognitive Causality.** A primary caveat of our analysis is the distinction between correlation and causation. While the Transition Advantage Matrix ($R$) identifies paths statistically over-represented in successful trajectories, it does not inherently prove that forcing a model into a "positive-advantage" state will yield a correct answer. For instance, a transition from Uncertainty to Backtracking is highly correlated with success in MATH-500; however, if the model lacks the underlying domain knowledge to rectify its error, steering it into a Backtracking state may merely result in "hallucinated corrections" or repetitive loops. Steering acts as a navigational guide, but it cannot substitute for the model's baseline knowledge engine. Investigating whether steering effectiveness is capped by the model's internal entropy or specific knowledge gaps is an interesting future work. Also, identifying a more robust and reliable reward signal beyond the empirical $R$ matrix remains a compelling future research direction.

**Limited Intervention Mechanism and Robustness.** We steer only at sentence boundaries using last-token embeddings, which can miss important mid-sentence decision points or fail when sentence segmentation is unclear. Moreover, linear activation steering (even with orthogonal injection) can have unintended side effects on content, style, or factuality, and its effectiveness depends on design choices such as the intervention layer and hyperparameters. We found that the effectiveness of steering varies depending on which layer of the model we intervene in. Currently, we identify the best layer through empirical testing on a hold-out set. A more principled or automated way to select these layers in real-time could make the method more robust across different model architectures.

**Sensitivity of Taxonomy.** We did not systematically evaluate alternative state granularities; however, we believe a coarser taxonomy (fewer states) would collapse functionally distinct behaviors, particularly *uncertainty estimation* and *backtracking* into broader categories, causing the Transition Advantage Matrix to lose resolution on precisely the metacognitive transitions most associated with success/failure. A finer taxonomy risks descriptive redundancy and data sparsity per transition, making reliable empirical estimation of $T(correct)$ and $T(incorrect)$ harder, which would destabilize the steering policy. Exploring alternative granularities is a promising future direction.

**Applicability to Open-Ended Tasks.** Computing the advantage matrix requires a binary success signal – a genuine limitation for open-ended tasks, but a fundamental challenge shared by the entire field, not specific to our framework. For such tasks, LLM-as-judge (Li et al., 2024) can serve as a surrogate signal: a strong judge (e.g., GPT-4.1) can label trajectories as high or low quality, replacing binary correctness. Rubric-based judging (Shen et al., 2026) has shown reliable enough signals for reward modeling even without ground truth.

## F. Diversity and Fluency Evaluation

We conducted an empirical evaluation using 100 random samples from MATH-500 dataset on *Qwen3-4B-Thinking*. We run 5 seeds per problem under both *default* and *Q-Value* steering conditions. We report the results in Table 6. We measured

*Table 6.* Diversity and fluency metrics under default and Q-Value steering conditions.

| Metric | Default | Q-Value Steering |
|---|---|---|
| Self-BLEU ↓ | $0.728 \pm 0.068$ | $0.727 \pm 0.064$ |
| Distinct-2 ↑ | $0.373 \pm 0.060$ | $0.372 \pm 0.057$ |
| Perplexity ↓ | $2.28 \pm 0.35$ | $2.29 \pm 0.35$ |

diversity via Self-BLEU and distinct-n (n=2), and fluency via perplexity (*Qwen2.5-3B* as the reference language model), reporting mean ± std. The results are nearly identical across all metrics, indicating no clear loss of diversity or fluency under Q-Value steering.

## G. Additional Results

To assess the generality of our FSM-based steering framework beyond the four primary models, we evaluate *DeepSeek-R1-Distill-Qwen-1.5B* on MATH-500 across all three steering variants. Results are reported in Table 7. Unlike the models in our main experiments, across all three steering conditions, accuracy remains essentially flat relative to the default baseline of 74.40%. Greedy and weighted steering produce marginal regressions (74.00% and 73.80%, respectively), while Q-Value steering yields the only positive movement, reaching 74.80%, a gain of 0.40 percentage points.

*Table 7.* Evaluation of DeepSeek-R1-Distill-Qwen-1.5B on MATH-500 across Default, Greedy, Weighted, and Q-Value steering variants. Metrics include Accuracy (%), average CoT generation length (Avg #Tokens), and average steering frequency (Avg #Steering). The best score for accuracy is shown in **bold**, and the second-best is underlined.

| Default | | Greedy | | | Weighted | | | Q-Value | | |
|---|---|---|---|---|---|---|---|---|---|---|
| Accuracy | Avg #Tokens | Accuracy ↑ | Avg #Tokens ↓ | Avg #Steering ↓ | Accuracy ↑ | Avg #Tokens ↓ | Avg #Steering ↓ | Accuracy ↑ | Avg #Tokens ↓ | Avg #Steering ↓ |
| 74.40 | 2773.73 | 74.00 | 3294.33 | 57.51 | 73.80 | 3019.22 | 22.66 | **74.80** | 2910.00 | 10.61 |

We attribute the limited impact to two compounding factors. First, distilled models compress a teacher's reasoning into dense, entangled representations; we assume that the cognitive states may not be as cleanly separable in activation space as in larger, natively-trained reasoners. As a result, the contrastive difference-of-means procedure may extract noisier transition steering vectors that reduces the precision of intervention. Second, at 1.5B parameters, the model's baseline accuracy of 74.40% may already approach the upper limit of what its internal knowledge and reasoning circuits can support on MATH-500. If the bottleneck is knowledge capacity rather than trajectory suboptimality, then even perfectly targeted steering cannot substitute for missing domain competence, a limitation which we discuss more broadly in Appendix E.

## H. Intervention Effectiveness

We provide a confusion matrix analysis in Table 8 for Q-Value steering on two models (GPT-L and QWEN), where the rows represent *baseline outcomes* (**BS = baseline success, BF = baseline fails**), and columns represent *steered outcomes* (**SS = steering success, SF = steering fails**). We consider **positive** = steering helped (intervention was useful) and **negative** = steering was not needed (baseline already correct).

This gives us a confusion matrix where:

- TN = both succeed (steering was redundant)
- FP = baseline succeeds but steering fails (steering was harmful)
- TP = baseline fails but steered succeeds (steering was helpful)
- FN = both fail (steering was ineffective)

This yields two key metrics:

- Rescue Rate $= TP/(TP + FN)$, measuring how often steering saved a failing trajectory.

- Harm Rate $= FP/(TN + FP)$, measuring how often steering broke a correct trajectory.

*Table 8.* Q-Value intervention effectiveness analysis on GPT-L and QWEN.

| Metrics | QWEN | | | | | | | | GPT-L | | | | | | | |
|---|---|---|---|---|---|---|---|---|---|---|---|---|---|---|---|---|
| | AIME | | MATH | | GPQA | | GSM8K | | AIME | | MATH | | GPQA | | GSM8K | |
| | SS | SF | SS | SF | SS | SF | SS | SF | SS | SF | SS | SF | SS | SF | SS | SF |
| BS | 24 | 1 | 453 | 2 | 106 | 17 | 912 | 127 | 11 | 2 | 385 | 10 | 96 | 18 | 1222 | 23 |
| BF | 2 | 3 | 4 | 41 | 19 | 56 | 134 | 146 | 6 | 11 | 31 | 74 | 27 | 57 | 26 | 48 |
| Rescue Rate | 40% | | 8.90% | | 25.30% | | 47.90% | | 35.30% | | 29.50% | | 32.10% | | 35.10% | |
| Harm Rate | 4.00% | | 0.40% | | 13.80% | | 12.20% | | 15.40% | | 2.50% | | 15.80% | | 1.80% | |

Across all settings, TN dominates, confirming that Q-value steering often leaves already-correct trajectories undisturbed. Rescue rates consistently exceed harm rates, showing that interventions on failing trajectories are effective. The low harm rates (0.4%–15.8%) validate that confidence-aware gating successfully avoids unnecessary disruption.

## I. Failure Analysis

**(a) Annotation.** We acknowledge several known edge cases: (1) mixed-function sentences that simultaneously *deduce* and express *uncertainty* (e.g., "So x = 5 but wait, that seems off"); (2) meta-commentary between *initialization* and *augmentation* (e.g., "This is a classic problem that can be solved with..."); (3) self-verification masquerading as *closure* (e.g., "So the answer is 42, let me double-check."); (4) epistemic hedging during *deduction* (e.g., "Assuming the definition..., f(x) is ... at x=0"). Our single-label-per-sentence annotation rule can oversimplify these cases, though they remain a minority.

**(b) Sentence Boundary Detection.** We have conducted diagnostic experiments on the MATH-500 benchmark. Across all four models, the overall false boundary rates are 84.73% (GPT-L), 84.90% (GPT-M), 88.70% (PHI), and 90.23% (QWEN). False boundary triggers are mostly caused by ellipsis tokens ("..."), accounting for 99.6% (GPT-L), 99.4% (GPT-M), and 99.6% (PHI) of false triggers, with QWEN at 93.2%. The remaining cases include relatively rare cases such as repeated exclamation marks, decimal points, in-formula periods, factorials and abbreviations. It is perhaps possible to augment the classifier to handle the most common of these cases.

The confidence gate largely mitigates this noise. Our gating mechanism (classifier confidence $\geq 90\%$) suppresses the majority of false boundary triggers: 35.6% (GPT-L), 56.0% (GPT-M), 80.0% (PHI), and 74.3% (QWEN). Slip-through rates (false boundaries that actually cause steering) remain low at 6.7%, 0.1%, 7.3%, and 18.9% respectively, demonstrating effective practical mitigation. The confidence gap confirms discriminability for most models. For GPT-L and QWEN, false boundaries elicit lower classifier confidence than true ones (gaps of $+0.057$ and $+0.007$). For GPT-M and PHI, the gap is slightly reversed ($-0.015$ and $-0.004$), meaning false boundaries elicit marginally higher classifier confidence, indicating that the confidence gate alone is insufficient in these cases and motivating future token-level granularity refinements as pursued by (Cui et al., 2025).

## J. Hyper-parameter Sensitivity

Our hyperparameter choices follow a principled methodology consistent with field norms, and layer search via held-out sets is also standard practice across the activation steering literature. Following (Chen et al., 2025a), which derives reasoning steering vectors from a subset of training samples, we select the best-performing layer on a held-out set once per model and fix it across all benchmarks. Hyper-parameters were found empirically following Chen et al. (2025a). We report a sensitivity study with different steering strength ($\alpha$) for weighted and Q-Value steering on *GPQA Diamond* benchmark in Table 9 below.

For each model, we include the selected $\alpha$ value as baseline in the paper, as well as two other $\alpha$ values for sensitivity analysis. In all cases, we select the $\alpha$ value that performs best.

*Table 9.* Sensitivity analysis of steering strength ($\alpha$) for Weighted and Q-Value Steering on GPQA Diamond across four models.

| Model | Weighted Steering | | Q-Value Steering | |
|---|---|---|---|---|
| | **Alpha** | **Accuracy (%)** | **Alpha** | **Accuracy (%)** |
| GPT-L | 0.10 | 59.60 | 0.50 | 59.09 |
| | 0.20 | 60.10 | 1.00 | 62.12 |
| | 0.50 | 58.59 | 1.20 | 61.62 |
| GPT-M | 0.20 | 61.62 | 0.50 | 64.14 |
| | 0.50 | 65.15 | 1.00 | 67.17 |
| | 1.00 | 64.14 | 1.20 | 66.67 |
| PHI | 0.20 | 70.71 | 0.20 | 66.67 |
| | 0.50 | 71.72 | 0.50 | 69.19 |
| | 1.00 | 69.70 | 1.00 | 65.15 |
| QWEN | 0.20 | 60.10 | 0.20 | 59.60 |
| | 0.50 | 62.12 | 0.50 | 63.13 |
| | 1.00 | 65.15 | 1.00 | 60.61 |

# K. Annotation Prompts

In this section, we provide the full prompt used for reasoning-state annotation with `GPT-4o-mini`.

```
### TASK
You are an expert reasoning-state annotator. Your task is to analyze a chain-of-thought
reasoning trace, broken into discrete text sentences, and label each sentence with one or more
*LABEL CODES* that describe what this sentence is *doing* functionally in the reasoning process.

### AVAILABLE LABELS and LABEL CODES
Each sentence may express one of the following tasks:
1. initialization (init) → Restating, reframing, or clarifying the task before starting reasoning.

2. deduction (deduce) → Performing calculations, making inferences, drawing implications, doing
computations, often doing intermediate conclusions.

3. augmentation (augment) → Strengthening or extending reasoning through any of the following:
- recalling facts, internal knowledge, provided information (augment-fact)
- stating or planning the solution (augment-plan)
- doing example testing or case trials (augment-test)
- exploring alternate solution paths (augment-branch)
- refining/correcting/verifying/merging previous steps, checking calculations, summarizing key
points, preparing final answer, re-confirming its own reasoning (augment-refine)
- using strategies not listed above (augment-emerge)

4. uncertainty estimation (uncertain) → Explicitly expressing doubt, confusion, hesitation, or lack
of confidence about current step, assumptions, calculations. May acknowledge missing information,
ambiguous problem framing, needs review or the possibility that deduction could be wrong.

5. backtracking (backtrack) → Revisiting earlier steps/assumptions, re-reading the instructions,
re-evaluating earlier results often following an uncertainty or realization of error.

6. final conclusion (closure) → Deciding a *final answer/option* to the query or output action, either
directly or as a meta-statement.

### ANNOTATION RULES
- Annotate the whole reasoning chain. Label each sentence separately.
- Match which definition best fits the sentence. Assign one label per sentence. Do NOT merge sentences.
- The first sentence may *NOT* always be a "init" state. It depends on the label definition.
- The last sentence is usually a "closure" state.
- "init" and "closure" should appear *at most once*.
- If a sentence in the *middle of a chain* expresses a final answer or meta-statement, label it as
"deduce" instead.
- Output only the labeled sentences, preserving exact original text: no text is omitted or paraphrased.
- Do not confuse *backtracking* with *uncertainty-estimation* or *augmentation*.
- Use the exact label codes: init, deduce, augment-subcategory, uncertain, backtrack, closure.
- Always format augmentation as: ["augment-fact"], ["augment-plan"], etc.

### OUTPUT FORMAT
For each sentence, wrap the text as:
["label"]
<sentence>
```

```
["end"]

### EXAMPLE
Input:
Multiply equation 2 with equation 3.Thus our earlier derivation leading to quartic equation is
correct. Simplifying the expression using substituition.Thus polynomial is correct. However, I'm not
fully sure. Let me read the question again. I will output the final answer now.

Expected Output:
["deduce"] Multiply equation 2 with equation 3. ["end"]
["deduce"] Thus our earlier derivation leading to quartic equation is correct. ["end"]
["deduce"] Simplifying the expression using substituition.["end"]
["deduce"] Thus polynomial is correct. ["end"]
["uncertain"] However, I'm not fully sure. ["end"]
["backtrack"] Let me read the question again. ["end"]
["closure"] I will output the final answer now. ["end"]

Now label each sentence of the given reasoning chain using the annotation rules.
No extra text, comments, or metadata.
```

## L. Prompt-Based Steering Templates

In this section, we provide the templates used for prompt-based steering for GPT-M on MATH-500 dataset.

**Positive Guidance**

When you reason, follow these rules:
1. Immediately after understanding or restating the problem: do deduction.
2. After deduction: if you are confident finish reasoning.
3. After you recall facts, plan the solution, do example testing, or explore alternate solution paths: do deduction.
4. If you feel uncertain or suspect a mistake: do deduction.
5. If you ever backtrack (reconsider earlier steps, re-read the question): do deduction.

**Negative Guidance**

When you reason, follow these rules:
1. Immediately after understanding or restating the problem: Do NOT recall facts, plan the solution, do example testing, or explore alternate solution paths.
2. After deduction: do NOT guess or suspect a mistake.
3. After you recall facts, plan the solution, do example testing, or explore alternate solution paths: do NOT guess or suspect a mistake.
4. If you feel uncertain or suspect a mistake: do NOT recall facts, plan the solution, do example testing, or explore alternate solution paths.
5. If you ever backtrack (reconsider earlier steps, re-read the question): Do NOT recall facts, plan the solution, do example testing, or explore alternate solution paths.

**Balanced Guidance**

When you reason, follow these rules:
1. Immediately after understanding or restating the problem:
(a) Do NOT recall facts, plan the solution, do example testing, or explore alternate solution paths.
(b) Do deduction.
2. After deduction:
(a) if you are confident finish reasoning.
(b) Do NOT guess or suspect a mistake.
3. After you recall facts, plan the solution, do example testing, or explore alternate solution paths:
(a) Do NOT guess or suspect a mistake.
(b) Do deduction.
4. If you feel uncertain or suspect a mistake:

(a) Do NOT recall facts, plan the solution, do example testing, or explore alternate solution paths.
(b) Do deduction.
5. If you ever backtrack (reconsider earlier steps, re-read the question):
(a) Do deduction.
(b) Do NOT recall facts, plan the solution, do example testing, or explore alternate solution paths.

**Cross-Model Guidance**

When you reason, follow these rules:
1. Immediately after understanding or restating the problem:
(a) Do NOT recall facts, plan the solution, do example testing, or explore alternate solution paths.
(b) Do deduction.
2. After deduction:
(a) guess or suspect a mistake.
(b) Do NOT recall facts, plan the solution, do example testing, or explore alternate solution paths.
3. After you recall facts, plan the solution, do example testing, or explore alternate solution paths:
(a) Do NOT guess or suspect a mistake.
(b) Do deduction.
4. If you feel uncertain or suspect a mistake:
(a) Do NOT recall facts, plan the solution, do example testing, or explore alternate solution paths.
(b) Do deduction.
5. If you ever backtrack (reconsider earlier steps, re-read the question):
(a) Do NOT recall facts, plan the solution, do example testing, or explore alternate solution paths.
(b) Do deduction.

# M. Example of Annotated CoT Reasoning

**Annotated Reasoning Chain (Sentence Level Annotation)**

["init"] We have points on line: A-B-C-D-E-F in order. ["end"]

["deduce"] We have distances along line: need coordinates. ["end"]

["augment-plan"] Let set positions along x-axis: let A at x=0. ["end"]

["deduce"] Then let AB = a, BC = b, CD = c, DE = d, EF = e. ["end"]
["deduce"] Then: ["end"]
["deduce"] AC = AB+BC = a+b =26 ...(1) ["end"]
["deduce"] BD = BC+CD = b+c =22 ...(2) ["end"]
["deduce"] CE = CD+DE = c+d =31 ...(3) ["end"]
["deduce"] DF = DE+EF = d+e =33 ...(4) ["end"]
["deduce"] AF = AB+BC+CD+DE+EF = a+b+c+d+e =73 ...(5) ["end"]
["deduce"] Also G has distances to C and D: CG = distance from C to G: suppose G has coordinates (xg, yg). ["end"]
["deduce"] Distance squared? given lengths: CG=40, DG=30. ["end"]

["augment-plan"] Also we need BGE area: use coordinates B and E on line (x positions) and G. ["end"]

["deduce"] We can find x positions of points. ["end"]
["deduce"] From equations, we solve a,b,c,d,e. ["end"]
["deduce"] We have a+b=26 ["end"]
["deduce"] b+c=22 ["end"]
["deduce"] c+d=31 ["end"]
["deduce"] d+e=33 ["end"]
["deduce"] Sum all: (a+b)+(b+c)+(c+d)+(d+e)=26+22+31+33=112 ["end"]
["deduce"] Left: a+2b+2c+2d+e =112. ["end"]
["deduce"] Also a+b+c+d+e=73 $\implies$ subtract: (a+2b+2c+2d+e)-(a+b+c+d+e)= b+c+d =112-73=39. ["end"]
["deduce"] So b+c+d =39. ["end"]
["deduce"] We know b+c=22, so d=39-22=17. ["end"]
["deduce"] Then c+d=31 $\implies$ c=14. ["end"]
["deduce"] Then b+c=22 $\implies$ b=8. ["end"]
["deduce"] a+b=26 $\implies$ a=18. ["end"]
["deduce"] Then d+e=33 $\implies$ e=16. ["end"]
["deduce"] Check AF sum: 18+8+14+17+16=73 OK. ["end"]

["augment-fact"] Now coordinates: let A at 0. ["end"]

["deduce"] Then B at a=18. ["end"]
["deduce"] C at a+b=26? Wait AC=26. yes C at 26. ["end"]
["deduce"] D at 26+c=40. ["end"]
["deduce"] E at 40+d=57. ["end"]
["deduce"] F at 57+e=73. ["end"]
["deduce"] So coordinates: B(18,0), C(26,0), D(40,0), E(57,0). ["end"]
["deduce"] Now G satisfies distances to C and D: $|CG| = 40, |DG| = 30$. ["end"]
["deduce"] So G is intersection of two circles centered at C(26,0) radius 40, and at D(40,0) radius 30. ["end"]

["augment-plan"] Compute coordinates. ["end"]

["deduce"] Let G=(x,y). ["end"]
["deduce"] Equations: ["end"]
["deduce"] $(x - 26)^2 + y^2 = 40^2 = 1600...(i)$ ["end"]
["deduce"] $(x - 40)^2 + y^2 = 30^2 = 900...(ii)$ ["end"]
["deduce"] Subtract (i)-(ii): $(x - 26)^2 - (x - 40)^2 = 700$. ["end"]
["deduce"] $Expand : (x^2 - 52x + 676) - (x^2 - 80x + 1600) = 700 \implies (-52x + 676 + 80x - 1600) = 700 \implies 28x - 924 = 700 \implies 28x = 1624 \implies x = 1624/28 = 58? \ compute \ 28 * 58 = 1624 \ yes. \ So \ x = 58.$ ["end"]
["deduce"] $Plug \ into \ (ii) : (58 - 40)^2 + y^2 = 900 \implies 18^2 + y^2 = 900 \implies 324 + y^2 = 900 \implies y^2 = 576 \implies y = 24.$ ["end"]

["uncertain"] Which sign? G not on line; area absolute uses magnitude. ["end"]

["deduce"] choose y=24 or -24 gives same area (absolute value). ["end"]
["deduce"] So G at (58,24). ["end"]
["deduce"] Check x=58 > E at 57 slightly right. ["end"]
["deduce"] Now need area of triangle BGE with points B(18,0), G(58,24), E(57,0). ["end"]

["augment-plan"] Compute using shoelace: coordinates: ["end"]

["deduce"] B(18,0) ["end"]
["deduce"] G(58,24) ["end"]
["deduce"] E(57,0) ["end"]
["deduce"] Area = 0.5 | x1*y2 + x2*y3 + x3*y1 - (y1*x2 + y2*x3 + y3*x1) | ["end"]
["deduce"] Compute: x1*y2 =18*24=432 ["end"]
["deduce"] x2*y3 =58*0=0 ["end"]
["deduce"] x3*y1 =57*0=0 ["end"]
["deduce"] Sum =432 ["end"]
["deduce"] Now y1*x2 =0*58=0 ["end"]
["deduce"] y2*x3 =24*57=1368 ["end"]
["deduce"] y3*x1 =0*18=0 ["end"]
["deduce"] Sum=1368 ["end"]
["deduce"] Difference:432-1368 = -936 abs 936. ["end"]
["deduce"] Area=0.5*936=468. ["end"]

["closure"] Thus answer 468. ["end"]

# N. Transition Advantage Matrices

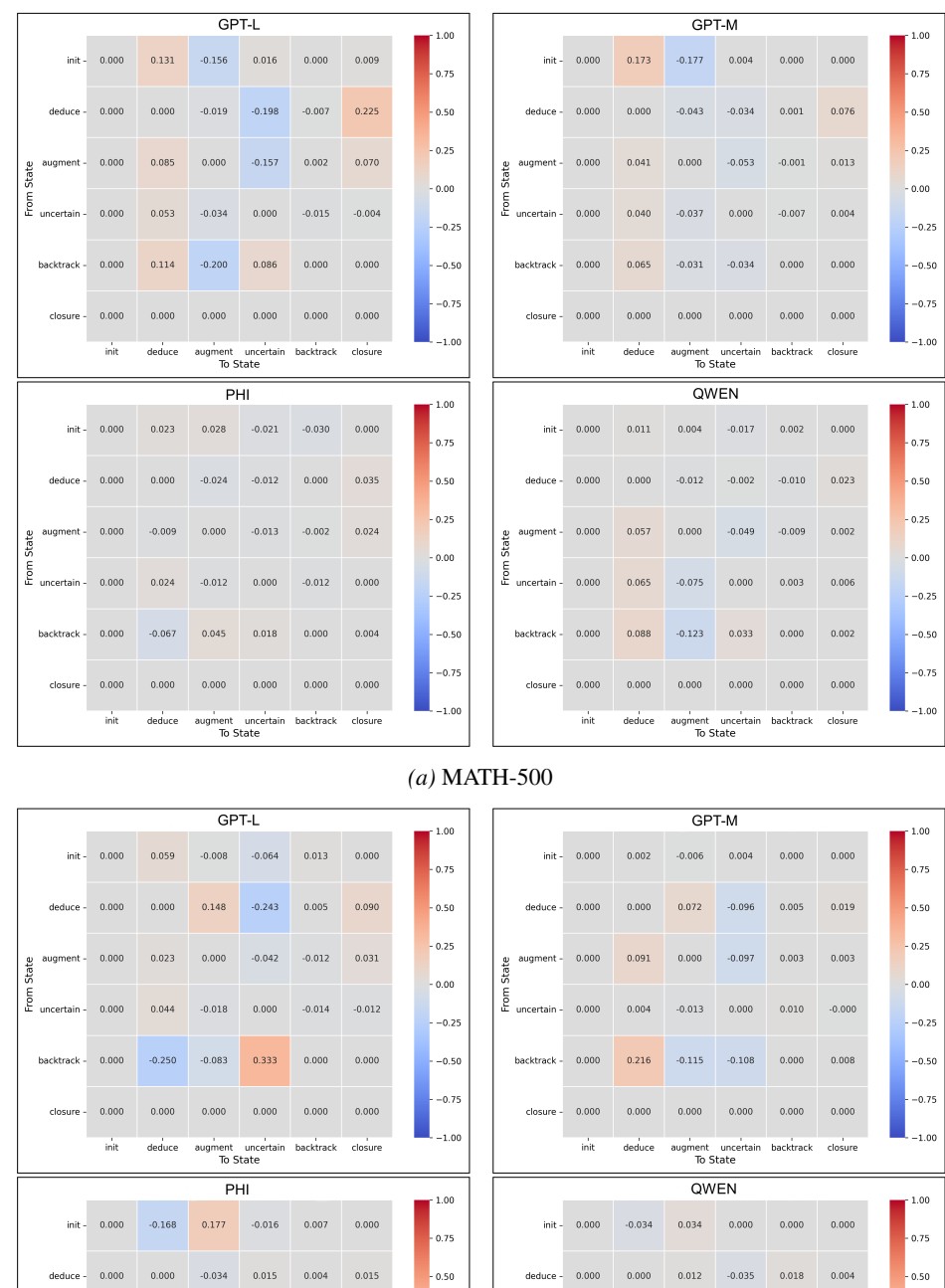

*(a)* MATH-500

*(b)* GPQA Diamond

*Figure 4.* Dataset-wise transition advantage matrices visualization across models.

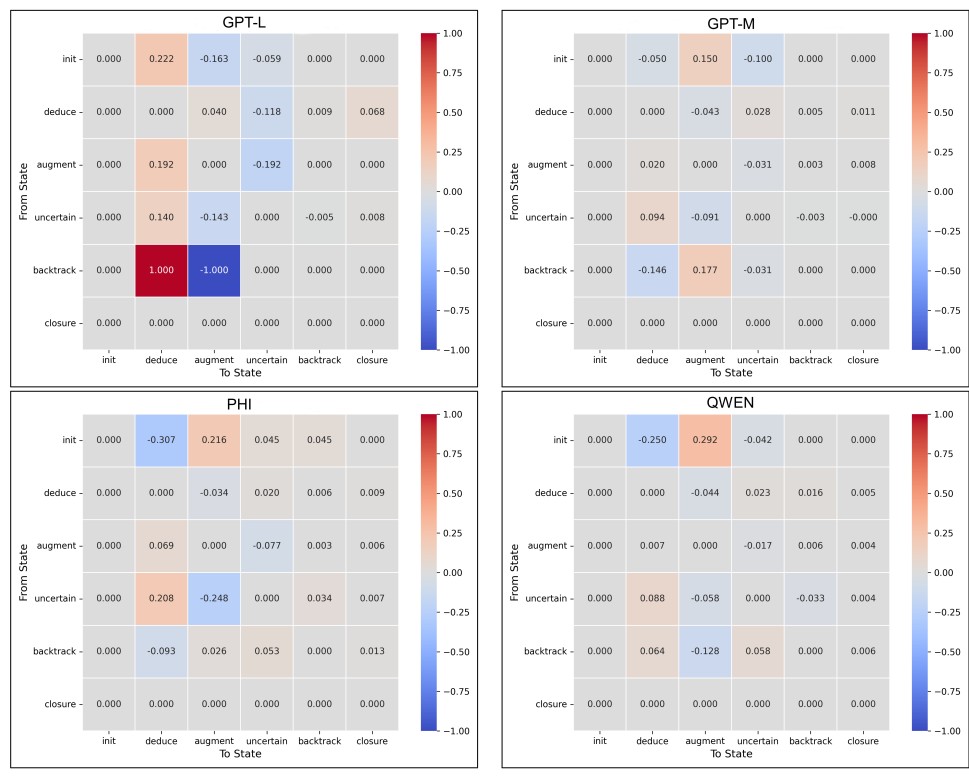

*(a)* AIME 25

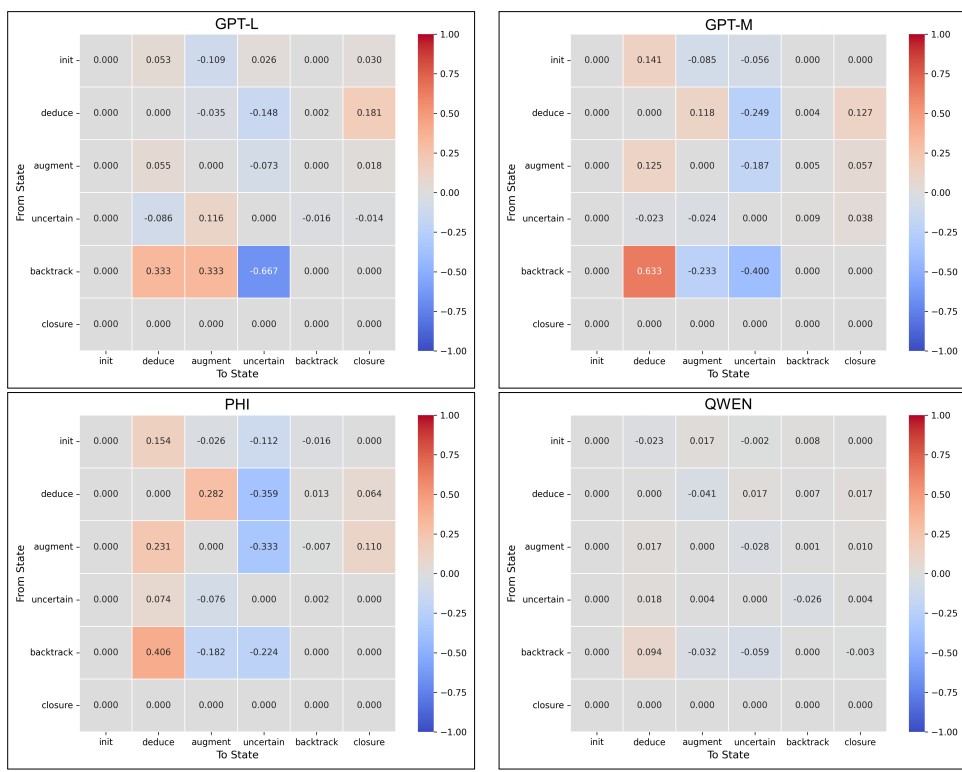

*(b)* GSM8K

*Figure 5.* Dataset-wise transition advantage matrices visualization across models.

