# OpenReview forum: "Modeling Hierarchical Thinking in Large Reasoning Models"
_ICML.cc/2026/Conference — ICML 2026 spotlight_

### Official Review · Reviewer_hYyH · 2026-02-27

**Soundness:** 3
**Presentation:** 4
**Significance:** 2
**Originality:** 3
**Overall Recommendation:** 4
**Confidence:** 4

**Summary:**

This paper models long Chain-of-Thought reasoning in large language models as transitions among a small set of discrete “cognitive states,” constructing a finite-state machine whose transition differences between correct and incorrect trajectories define a transition advantage signal. The authors use this signal within a Q-value iteration framework to plan sparse, inference-time activation steering interventions at sentence boundaries, nudging hidden states toward more successful reasoning transitions without updating model weights. Experiments on mathematical and structured reasoning benchmarks show modest accuracy improvements and substantially fewer interventions compared to greedy or heuristic steering, positioning the method as a structured, planning-based approach to controllable reasoning.

**Compliance With Llm Reviewing Policy:**

Affirmed.

**Final Justification:**

While the scope still fails limited and artificial on the current benchmarks (they are deduction heavy, whereas real world problems are abduction and induction heavy), other concerns are resolved. Rating +1.

**Key Questions For Authors:**

Basically same as weakness.

Some other questions are:
1) How did you developed the 6 states? Why do you think they are sufficient? Why are they generalizable?
2) Did you mannually verify the gpt-4o-mini (a weak model)'s annotation? Does it really make sense? What are corner cases that are outside the 6?
3) For a very ill-defined task like deep-research, without verifiable ground truth, how could you still apply this method - since the Rij will not be easily calucable? Maybe this is my bias but math problems alone cannot convince people the significance.

**Limitations:**

yes

**Strengths And Weaknesses:**

## Strengths

**S1.** The paper proposes a structured framework that models long Chain-of-Thought (CoT) reasoning as transitions among a small set of discrete “cognitive states,” and further leverages differences between successful and failed trajectories to construct a transition advantage signal. The integration of value iteration (Q-value planning) to enable long-horizon, sparse activation steering is conceptually elegant and more principled than purely greedy or heuristic steering strategies. The idea of treating inference-time control as a planning problem is novel within the activation-steering literature.

**S2.** The work attempts to bridge interpretability and control. The same finite-state abstraction is used both to analyze reasoning structure and to derive actionable steering signals. This unification is conceptually appealing and distinguishes the work from prior activation-patching methods that focus only on local behavior manipulation.

---

## Weaknesses

**W1. Limited task diversity and weak generalization evidence.**
The empirical evaluation is heavily concentrated on mathematical reasoning benchmarks, with only marginal extension to related structured reasoning datasets. Mathematical problem solving is highly stylized, typically single-answer, and often exhibits clean phase transitions in reasoning (e.g., deduction, uncertainty, correction). It is unclear whether the proposed state abstraction and transition advantage modeling generalize to broader reasoning domains such as multi-hop retrieval, logical games, symbolic manipulation, planning, code synthesis, or open-ended multi-constraint reasoning. Without evaluation across qualitatively different reasoning tasks, it is difficult to assess whether the method captures a general property of hierarchical reasoning or merely exploits patterns specific to math-style CoT generation.

**W2. Unclear positioning between method paper and interpretability paper.**
The paper occupies an ambiguous position. On one hand, it argues for effectiveness primarily through downstream performance improvements, suggesting a methodological contribution. On the other hand, it frames the finite-state abstraction as an interpretability insight into hierarchical thinking. However, as a method paper, it lacks comparison against strong external reasoning baselines such as self-consistency, multi-sample reranking, verifier-based filtering, reflection prompting, or lightweight learned critics. Most comparisons are internal variants (greedy vs weighted vs Q-value) or prompt-based adaptations derived from the same transition matrix. As an interpretability paper, the central insight—six discrete reasoning states governed by first-order transitions—is not independently validated beyond its utility for steering. This dual positioning weakens the clarity of contribution and evaluation criteria.

**W3. Strong Markov assumption and insufficient modeling of long-term dependency.**
The core formulation assumes a first-order Markov structure over reasoning states, i.e., the next state depends only on the current state. In practice, reasoning trajectories often exhibit higher-order dependencies: repeated failures, accumulated inconsistencies, or multiple unsuccessful augmentations may change the optimal future transition in ways that cannot be captured by a one-step transition matrix. Although Q-value iteration introduces long-horizon planning over immediate rewards, the reward signal itself is derived from first-order transition statistics. The paper does not explore higher-order transition models, history-conditioned policies, or neural approximations of the advantage/Q function. As a result, it remains unclear whether the first-order abstraction is an empirically sufficient modeling choice or merely a simplifying assumption adopted for tractability.

**W4. The “training-free” and “plug-and-play” framing is somewhat misleading.**
While the base model parameters are not updated, the method requires substantial offline computation: generating reasoning traces, labeling sentence-level states (with an external large model), separating correct and incorrect trajectories, estimating transition statistics, and extracting steering directions. This process effectively constructs a learned control layer prior to inference-time use. If the task distribution, prompting style, or model variant changes, these statistics may need to be recomputed. Thus, the approach is not strictly off-the-shelf inference-time control but rather offline-estimated control with lightweight online application. The computational and practical implications of this distinction are not sufficiently discussed.

**W5. Limited statistical rigor and modest effect sizes.**
Several reported improvements are small (often within 1–2 percentage points), particularly for smaller models. On small benchmarks (e.g., AIME-style datasets), such differences may correspond to only one or two examples. The paper does not report confidence intervals, significance tests, or multi-seed variance. Without stronger statistical validation, it is difficult to determine whether the improvements are robust. In many cases, the most convincing benefit appears to be reduced intervention frequency rather than large accuracy gains; however, this shift in emphasis is not clearly articulated in the main narrative.

**W6. Potential confound between linguistic style and “cognitive state.”**
The six reasoning states are defined at the sentence level and labeled via an external model. It is plausible that these states correlate strongly with surface linguistic cues (e.g., expressions of uncertainty, explicit corrections, or structured deduction markers) rather than latent cognitive modes. If so, the transition matrix may reflect stylistic or rhetorical patterns rather than deeper reasoning structure. The paper does not provide causal analyses (e.g., style-controlled paraphrases) to disentangle linguistic form from reasoning function, which weakens the interpretability claim.

---

> ### Author Rebuttal · Authors · 2026-03-31
>
> Thank you for your thoughtful comments. Please see our responses below.
>
> **Note: References are in our response to *Reviewer URnh* to save space.**
>
> ---
>
> **Q1) Task Diversity and Generalization Evidence**: Math benchmarks are the standard testbed for LRM reasoning papers [7, 9, 10], which follow the same approach before broader generalization. Starting with verifiable tasks is a principled methodological choice, not a fundamental limitation of the framework itself. Our framework already goes beyond math: GPQA Diamond covers graduate-level physics, chemistry, and biology, requiring multi-hop scientific reasoning. Adapting it to code generation or open-ended reasoning may require additional states (e.g., distinguishing code execution from planning).  Thank you for this suggestion which provides a concrete and promising future direction.
>
> ---
>
> **Q2) Clarifying Paper Position**: Following [7, 11], we utilize a well-established paradigm: **abstraction first, application second**. Our FSM abstraction is designed to capture the reasoning progression; activation steering is one demonstration of its utility. Combining steering with self-consistency reranking is a promising future direction. Thank you for this excellent suggestion, which we will include if the paper is accepted.
>
> Beyond steering utility, the FSM abstraction independently reveals meaningful structure in LRM reasoning: (1) Cohen's κ=0.89 confirms the states are human-distinguishable; (2) the advantage matrix reveals significant structural differences between correct/incorrect trajectories; (3) >90% classifier accuracy confirms the states are encoded in hidden representations, not surface text patterns. If the states were arbitrary, none of these would hold, constituting interpretability validation independent of downstream utility.
>
> ---
>
> **Q3) Markov Assumption**: Please see our response under **Q1** of **Reviewer URnh**.
>
> ---
>
> **Q4) Training-free Framing**: Please see our response under **Q4** of **Reviewer RUUK**.
>
> ---
>
> **Q5) Statistical Validation**: Within the rebuttal window, we ran a multi-seed experiment  on *MATH-500/GPT-L*. We sampled 100 problems randomly and ran 15 seeds per problem for both default and Q-value steering. Per-problem accuracy was estimated as the mean of 15 binary outcomes and the reported accuracy is the mean of these 100 per-problem estimates. On this subset, default and steered accuracies are 78.92% and 82.01%, corresponding to a mean improvement of +3.09 percentage points (pp) with (SE: 1.07pp). The 95% standard CI is [+0.98, +5.20] pp, and bootstrap resampling over the 100 per-problem deltas (10,000 iterations) yields a 95% CI of [+0.68, +5.31] pp.
>
> ---
>
> **Q6) Linguistic Style vs. Cognitive State Confound Clarification**: Our states are validated in latent space, not just text. State classifiers trained on hidden representations achieves >90% accuracy across all settings -- purely stylistic states would not be reliably distinguishable. This aligns with Coconut [12], which shows reasoning states are meaningfully encoded in the LLM's last hidden state, distinct from surface tokens. Our activation steering provides causal evidence: intervening directly on hidden states at sentence boundaries with no surface text modification consistently improves task accuracy across 4 benchmarks and 3 models. If states only captured rhetorical patterns, steering vectors derived from them would have no systematic effect on reasoning outcomes. We agree that style-controlled paraphrase experiments would further strengthen interpretability claims and will explore this in future work.
>
> ---
>
> **Q7) Generalizability of States**: Please see our response under **Q1** of **Reviewer RUUK**.
>
> ---
>
> **Q8) Annotation Failure Cases**: We reported annotation verification in Section 4. We acknowledge several known edge cases: (1) mixed-function sentences that simultaneously deduce and express uncertainty (e.g., "So x = 5 but wait, that seems off"); (2) meta-commentary between initialization and augmentation (e.g., "This is a classic problem that can be solved with..."); (3) self-verification masquerading as closure (e.g., "So the answer is 42... let me double-check"); (4) epistemic hedging during deduction (e.g., "Assuming the definition..., f(x) is ... at x=0"). Our single-label-per-sentence annotation rule can oversimplify these cases, though they remain a minority. An explicit corner case discussion will be added in the final version.
>
> ---
>
> **Q9) Applicability to Open-Ended Tasks**: Computing advantage matrix requires a binary success signal, a limitation for open-ended tasks, but a fundamental challenge shared by the entire field. For such tasks, LLM-as-judge [13] can serve as a surrogate signal: a strong judge (e.g., GPT-4o) labels trajectories as high/low quality, replacing binary correctness. Rubric-based judging [14] has shown reliable enough signals for reward modeling even without ground truth. We will explicitly discuss this in the final version.

---

> > ### Author Rebuttal · Reviewer_hYyH · 2026-04-01
> >
> > While the scope still feels kind of limited and artificial on the current benchmarks (they are deduction heavy, whereas real world problems are abduction and induction heavy), other concerns are resolved. Rating +1.

---

> > > ### Author Response · Authors · 2026-04-02
> > >
> > > Thank you so much for updating your score. We are pleased to hear that most of your concerns have been resolved. We are happy to address any further questions you may have.

---

### Official Review · Reviewer_URnh · 2026-02-27

**Soundness:** 3
**Presentation:** 3
**Significance:** 3
**Originality:** 4
**Overall Recommendation:** 5
**Confidence:** 4

**Summary:**

This study proposes a training-free inference-time control method that treats reasoning as a topology of reasoning type transitions. It estimate the long-horizon utility of state transitions and apply sparse, orthogonal activation steering at sentence boundaries to align the CoT generation with optimal reasoning policies.

**Compliance With Llm Reviewing Policy:**

Affirmed.

**Final Justification:**

My concerns have been adequately addressed.
Thank you for the rebuttal. I will maintain my score.

**Key Questions For Authors:**

See "Strengths And Weaknesses".

**Limitations:**

yes

**Strengths And Weaknesses:**

- Strength
  - This study sees the reasoning as a topology of reasoning type, and based on the topology graph, it accelerates the correct transition of reasoning by utilizing steering vectors and Q-value which is constructed from the transition map. This approach is novel and interesting (personally exciting).
  - Reasoning analysis/interpretability/solution is important for recent LRMs; this study integrates these aspects with new insights and findings.
  - A comprehensive experiment is conducted with a variety of models and datasets, demonstrating the effectiveness of the proposed method.
  - The paper is well-written and easy to understand.
- Weakness
  - The transition map for the proposed method is based on only the previous state (t-1), but actually it depends on the history of all the transitions (1, ..., t-1).
  - The proposed approach requires a lot of hyperparameter settings or conditional rules, which might be a practical obstacle for practitioners or developers.
    - How about the sensitivity/robustness to the performance?
  - There is already some prior work that treats reasoning as graphs (but they don't propose an improvement method; just analysis). But I think at least authors should cite them at related work section.
    - https://arxiv.org/abs/2506.05744
    - https://arxiv.org/abs/2505.13890
    - https://arxiv.org/abs/2509.21128
  - 3.4: The definition of the orthogonal vector seems to be incorrect. I'm wondering if it is just a mistake in the description or if the authors need to rerun the experiment.

---

> ### Author Rebuttal · Authors · 2026-03-31
>
> We're glad you found our work interesting and impactful. We address your comments below:
>
> ---
>
> **Q1) Markov Assumption**: We acknowledge that the Markov assumption is a first order approximation because it does not capture the state of the model.  We considered memory-based models such as push-down automata but discounted them because the context of a model is significantly different from a stack.  We hope to explore more sophisticated models in our future work.
>
> To empirically validate the impact of the Markovian assumption, we conducted a second-order analysis on MATH-500 (Qwen3-4B-Thinking):
>
> (1) computing first and second-order transition matrices for correct/incorrect trajectories,
>
> (2) deriving $R$ and $R^{2}$, then comparing per-triplet sign agreement for every valid (prev, curr, next),
>
> (3) checking *sign($R$[curr,next])* == *sign($R^{2}$[prev→curr, next])*.
>
> **Sign Agreement: 82.1% (64/78 triplets)** – first-order $R$ correctly identifies beneficial transition direction in 4/5 contexts. The 14 disagreements concentrate on near-zero $R$ values (mean $|R|≈0.01$) – neutral transitions where steering is not triggered. **Pearson r=0.63, Spearman r=0.62** (p<1e-9) confirm $R$ and $R^{2}$ are strongly correlated. **Mean entropy ΔH=0.019 bits** – second-order adds negligible predictive information beyond first-order.
>
> Due to response length limitations, we report analysis on one model/dataset here; full analysis across all settings will be included in the final version.
>
> ---
>
> **Q2) Hyper-parameter Sensitivity**: Thank you for the suggestion.  We want to clarify two points: (1) our hyperparameter choices follow a principled methodology consistent with field norms, and (2) layer search via held-out sets is also standard practice across the activation steering literature. Following [9], which derives reasoning steering vectors from a subset of training samples, we select the best-performing layer on a held-out set once per model and fix it across all benchmarks. To summarize, hyperparameters were found empirically following [9].
>
> Due to character constraints, we report a sensitivity study with different steering strength (alpha) for weighted and Q-value iteration on *GPQA Diamond* in the table below.
>
> | Model |   Weighted Steering | |   Q-Value Steering | |
> |---|---|---|---|---|
> | | Alpha | Accuracy (%) | Alpha | Accuracy (%) |
> | | 0.10 | 59.60 | 0.50 | 59.09 |
> | GPT-L | 0.20 | 60.10 | 1.00 | 62.12 |
> | | 0.50 | 58.59 | 1.20 | 61.62 |
> ||||||
> | | 0.20 | 61.62 | 0.50 | 64.14 |
> | GPT-M | 0.50 | 65.15 | 1.00 | 67.17 |
> | | 1.00 | 64.14 | 1.20 | 66.67 |
> ||||||
> | | 0.20 | 70.71 | 0.20 | 66.67 |
> | PHI | 0.50 | 71.72 | 0.50 | 69.19 |
> | | 1.00 | 69.70 | 1.00 | 65.15 |
> ||||||
> | | 0.20 | 60.10 | 0.20 | 59.60 |
> | QWEN | 0.50 | 62.12 | 0.50 | 63.13 |
> | | 1.00 | 65.15 | 1.00 | 60.61 |
>
> For each model, we include the selected alpha value as baseline in the paper, as well as two other alpha values for sensitivity analysis.  In all cases, the selected alpha value performs best.  We will include an appendix section with a more comprehensive sensitivity analysis of hyperparameters across all settings in the final version.
>
> ---
>
> **Q3) Adding Prior Works**: Thank you for the suggestions which we will include if the paper is accepted.
>
> ---
>
> **Q4) Notation Correction**:  Thank you for pointing this out. This is a typographical error in the description which we will correct. The formula should read:
>
> $\hat{h} = \frac{h}{|h|_2 + \epsilon}$
>
> $v_{\perp} = v - (v^{\top}\hat{h})\hat{h}$
>
> The implementation uses the correct formula.
>
> ---
>
> **References**
>
> [1] Alan H. Schoenfeld. Mathematical Problem Solving. Academic Press 1985
>
> [2] Nelson & Narens. Metamemory: A theoretical framework and new findings, Academic Press 1990
>
> [3] Bogdan et al. Thought Anchors: Which LLM Reasoning Steps Matter?, ArXiv 2025
>
> [4] Chauhan et al. Punctuations and Predicates in Language Models, EACL 2026
>
> [5] Turner et al. Steering language models with activation engineering, ArXiv 2024
>
> [6] Arditi et al. Refusal in language models is mediated by a single direction, NeurIPS 2024
>
> [7] Venhoff et al. Understanding reasoning in thinking language models via steering vectors, ICLR 2025 Workshop
>
> [8] Zou et al. Representation engineering: A top-down approach to ai transparency, ArXiv 2023
>
> [9] Chen et al. Seal: Steerable reasoning calibration of large language models for free, COLM 2025
>
> [10] Guo et al. Deepseek-r1: Incentivizing reasoning capability in llms via reinforcement learning, ArXiv 2025
>
> [11] Besta et al. Graph of thoughts: Solving elaborate problems with large language models, AAAI 2024
>
> [12] Hao et al. Training large language models to reason in a continuous latent space, COLM 2025
>
> [13] Li et al. Llms-as-judges: a comprehensive survey on llm-based evaluation methods, ArXiv 2024
>
> [14] Shen et al. Rethinking Rubric Generation for Improving LLM Judge and Reward Modeling for Open-ended Tasks, ArXiv 2026

---

> > ### Author Rebuttal · Reviewer_URnh · 2026-04-02
> >
> > Thank you for the rebuttal. I will maintain my score.

---

> > > ### Author Response · Authors · 2026-04-02
> > >
> > > We appreciate your comments and we are pleased to hear that your concerns have been resolved. We are happy to address any further questions you may have.

---

### Official Review · Reviewer_gRWM · 2026-03-08

**Soundness:** 3
**Presentation:** 3
**Significance:** 3
**Originality:** 3
**Overall Recommendation:** 5
**Confidence:** 4

**Summary:**

This paper explores a way of modeling the reasoning traces of large reasoning models. They model these traces using a finite state machine with six transitions. They also propose an inference-time Q-Value guided steering mechanism that improves performance of three models on four common benchmarks.

**Compliance With Llm Reviewing Policy:**

Affirmed.

**Final Justification:**

Rebuttal addressed points.

**Key Questions For Authors:**

1. Is it possible to include more models in your analysis?
2. Can you provide an analysis of when the steering policy intervenes correctly versus unnecessarily?
3. How sensitive are the results to the choice of the six-state taxonomy? Have you tested alternative granularities (ie. fewer or more states), and if so, does the transition advantage matrix and resulting steering policy remain stable?

**Limitations:**

yes

**Strengths And Weaknesses:**

**Soundness:**

Soundness strengths:

- Good to include multiple baselines for evaluation in the experimental results section.
- The Q-value steering approach was the most effective across almost all models and baselines.
- The paper evaluates the method across several benchmarks with different difficulty profiles, which helps show that the proposed approach is not tied to only one narrow task setting.
- I appreciated that the authors compare not just accuracy, but also token usage and intervention frequency; this gives a more complete empirical picture of the tradeoff between effectiveness and efficiency.

I have the following soundness concerns:

- Can you include a table with a confusion matrix that compares the TP, FP, TN, FN statistics for the application of your intervention method? For example, how many trajectories stopped (or intervened on) were actually incorrect? How many trajectories not stopped (or intervened on) were incorrect? If you can compare those percentages with the base rates, that would be helpful for readers.





**Presentation:**

The paper had the following presentation strengths:

- It was good to acknowledge that finite state machines do not capture memory, which might have had the ability to better model your reasoning dynamics. This leaves the door open for future work.
- The formalization section was intuitive and easy to follow.
- Figure 2 is helpful conceptually: even though some text is difficult to read, the figure does communicate the end-to-end pipeline from annotation to control.

Here are a few of my concerns:

- I caution against anthropomorphic language like “hierarchical thinking” and “cognitive phases”, used across the paper and in the title. The intervention that you propose reduces error, but doesn’t provide evidence that these models are ‘thinking’ in the way that we might expect. I would reduce vocabulary to only the needed language to describe your approach and connection to the literature.
- The text was too small in Figure 2. I could not read the tables even when zoomed 500% on my laptop.
- In section 3.2, I found the term “capability” confusing. It might be more intuitive to decouple steering from control, instead.
- On the bottom of page 6, what is meant by “cognitive depth”?
- “Related Works” should be changed to “Related Work”.




Significance:
Significance strengths:

- The paper addresses a meaningful problem: if reasoning traces can be modeled at a higher level of abstraction, this could improve both interpretability and test-time control for reasoning models.
- I think the intervention-efficiency angle is potentially important. Reducing the number of steering actions while preserving or improving accuracy could matter for practical deployment.

Some significance weaknesses that I noticed include:

- I am wondering why you only tested three models on your method. Can you extend this to a greater suite, including current state of the art models like the GPT-5 series? Most benchmarking papers include GPT, Claude, and Gemini SOTA models. It would be good to show that you method works here, too.
- Your intervention method involves annotating every sentence to a state. This is computationally very expensive, as it increases the number of inference calls needed by the number of sentences in the reasoning trace. Can you include an analysis of the distribution of reasoning strength lengths in your application? Can you clarify in the paper that no extra annotation steps are needed at inference-time to use your intervention method? How many total annotations were collected?
- I wonder how much improvement you’d get by applying your pipeline independently for each model, to better capture between-model differences. Is this information available?




**Originality:**

Originality strengths include:

- Previous interventions adjusted or flagged reasoning outputs by using local methods and heuristics, but this paper includes a method to establish a global hierarchical view of how reasoning traces evolve in a model.
- They use a clever trick the create a planning-aware steering policy: mapping free-form reasoning traces into a finite state machine with 6 transitions. They can study the transition structure to compare correct and incorrect outputs and use this information to construct their intervention.
- They derive a long-horizon CoT control policy using Q-value iteration.

---

> ### Author Rebuttal · Authors · 2026-03-31
>
> We thank you for your suggestions and questions. Please see our responses below.
>
> **Note: References are in our response to *Reviewer URnh* to optimize space.**
>
> ---
>
> **Q1) Intervention Effectiveness**: We provide a confusion matrix analysis in Table 1 for Q-Value steering on two models, where the rows represent *baseline outcomes* **(BS = baseline success, BF = baseline fails)**, and columns represent *steered outcomes* **(SS = steer success, SF = steer fails)**.
>
> We consider:
>
> - positive = steering helped (intervention was useful)
>
> - negative = steering was not needed (baseline already correct)
>
> This gives us:
>
> - TN (A) = both succeed, steering was redundant
>
> - FP (B) = baseline succeeds but steered fails, steering was harmful
>
> - TP (C) = baseline fails but steered succeeds, steering was helpful
>
> - FN (D) = both fail, steering was ineffective
>
> This yields two key metrics:
>
> - Rescue Rate = C/(C+D), measuring how often steering saved a failing trajectory
>
> - Harm Rate = B/(A+B), measuring how often steering broke a correct trajectory
>
> | | | | | QWEN | | | | | | | | | GPT-L | | | | | |
> |---|---|---|---|---|---|---|---|---|---|---|---|---|---|---|---|---|---|---|
> | | AIME | | MATH | | GPQA | | GSM8K | | | AIME | | MATH | | GPQA | | GSM8K | |
> | | SS | SF | SS | SF | SS | SF | SS | SF | | SS | SF | SS | SF | SS | SF | SS | SF |
> | BS | 24 | 1 | 453 | 2 | 106 | 17 | 912 | 127 | | 11 | 2 | 385 | 10 | 96 | 18 | 1222 | 23 |
> | BF | 2 | 3 | 4 | 41 | 19 | 56 | 134 | 146 | | 6 | 11 | 31 | 74 | 27 | 57 | 26 | 48 |
> | Rescue Rate | 40% | | 8.90% | | 25.30% | | 47.90% | | | 35.30% | | 29.50% | | 32.10% | | 35.10% | |
> | Harm Rate | 4.00% | | 0.40% | | 13.80% | | 12.20% | | | 15.40% | | 2.50% | | 15.80% | | 1.80% | |
>
> Table 1: Intervention Effectivness Analysis
>
> Across all settings, A (TN) dominates, confirming that steering most often leaves already-correct trajectories undisturbed. Rescue rates consistently exceed harm rates, showing that interventions on failing trajectories are effective. The low harm rates (0.4%–15.8%) validate that confidence-aware gating successfully avoids unnecessary disruption. We will include this analysis in the final version.
>
> ---
>
> **Q2) Presentation Suggestions**: We agree and will reduce anthropomorphic language (e.g., replacing "hierarchical thinking" with "structured reasoning") and revise the title accordingly. Figure 2 texts will be enlarged for readability. In Section 3.2, we will decouple "steering mechanism" from "control logic" explicitly. "Cognitive depth" will be replaced with "reasoning complexity", and "Related Works" will be corrected to "Related Work."
>
> ---
>
> **Q3) Annotation and White-box Clarification**: We evaluated open-weight reasoning models because our method requires white-box access to hidden activations for steering vector extraction, which is unavailable for closed-source models like GPT-5, Claude, or Gemini. This is not a limitation of our approach but a constraint shared by all activation steering works in the literature [5 - 9].
>
> We want to clarify that no annotation calls are made at inference time. Following [7], we used GPT-4o-mini for annotation, which is used only during the offline training phase to label CoT trajectories for (1) computing the advantage matrix and (2) training the state encoder and classifiers. At inference time, state classification is performed entirely by our lightweight trained MLP classifiers (Section 3.2), which operate directly on the model's hidden representations with negligible overhead.
>
> Our pipeline works independently per model. Each model has its own transition matrices, steering vectors, and classifiers trained on its own hidden representations. Table 1 results already reflect this per-model specialization.
>
> ---
>
> **Q4) Additional Results**: We evaluated an additional white-box model, *DeepSeek-R1-Distill-Qwen-1.5B*, on MATH-500. All three steering variants are evaluated against the default baseline (74.40% accuracy, 2773.73 avg tokens). Greedy achieves 74.00% accuracy (3294.33 avg tokens, 57.51 avg steering), Weighted achieves 73.80% (3019.22 avg tokens, 22.66 avg steering), and Q-Value achieves 74.80% (2910 avg tokens, 10.61 avg steering). Complete results for all three steering variants across other benchmarks will be reported in the final version.
>
> ---
>
> **Q5) Sensitivity of Taxonomy**: We did not systematically evaluate alternative state granularities; however, we believe a coarser taxonomy (fewer states) would collapse functionally distinct behaviors, particularly Uncertainty and Backtracking into broader categories, causing the Transition Advantage Matrix to lose resolution on precisely the metacognitive transitions most associated with success/failure. A finer taxonomy risks descriptive redundancy and data sparsity per transition, making reliable empirical estimation of T(correct) and T(incorrect) harder, which would destabilize the steering policy. Exploring alternative granularities is a promising future direction.

---

> > ### Author Rebuttal · Reviewer_gRWM · 2026-04-01
> >
> > My concerns have been fully resolved and I will bump by score up by 1 to reflect this.

---

> > > ### Author Response · Authors · 2026-04-01
> > >
> > > Thank you for updating your score and we are glad to hear that your concerns have been resolved. Please let us know if you have any further questions or comments. We would be happy to provide additional clarifications.

---

### Official Review · Reviewer_RUUK · 2026-03-12

**Soundness:** 3
**Presentation:** 4
**Significance:** 3
**Originality:** 2
**Overall Recommendation:** 5
**Confidence:** 4

**Summary:**

This paper proposes modeling the Chain-of-Thought (CoT) reasoning process of Large Reasoning Models through a Finite State Machine (FSM). Based on the predefined six abstract cognitive states: Initialization, Deduction, Augmentation Strategy, Uncertainty Estimation, Backtracking, and Final Conclusion, each sentence in a CoT is automatically labeled into one of these states using GPT-4o-mini. The authors then compute outcome-conditioned transition matrices to derive a Transition Advantage Matrix $R$, which captures which types of state transition are statistically represented in successful versus failed reasoning trajectories.
Building on this, the paper introduces three steering method, Greedy Selection, Q-Value Iteration, and Weighted Steering. At sentence boundaries during inference, the model's current and predicted next states are classified from hidden representation, and if the predicted transition is suboptimal according to the advantage matrix $R$, an orthogonal activation steering vector is injected to redirect the reasoning. Experiments across four benchmarks (AIME25, MATH-500, GPQA Diamond, GSM8K) and three open reasoning models (Qwen3-4B-Thinking, gpt-oss-20b, Phi-4-reasoning) demonstrate that Q-Value steering achieves consistent accuracy improvements, often requiring dramatically fewer interventions than greedy or weighted baselines.

**Compliance With Llm Reviewing Policy:**

Affirmed.

**Final Justification:**

This paper proposes modeling the Chain-of-Thought (CoT) reasoning process of Large Reasoning Models through a Finite State Machine (FSM) framework. Based on the predefined six abstract cognitive states, Initialization, Deduction, Augmentation Strategy, Uncertainty Estimation, Backtracking, and Final Conclusion, the approach automatically labels each sentence in a CoT sequence using GPT-4o-mini. Those states contribute to compute outcome-conditioned transition matrices for successful reasoning trajectories. The technical pipeline is well-constructed, and the FSM formalization along with the overall pipeline design is intuitive and clearly motivated.
My primary concern centered on the rigidity of the underlying assumptions, particularly the reliance on predefined states and the Markov assumption. The authors addressed these concerns during the rebuttal phase by providing additional justification for the state taxonomy, discussing its generalizability, and including a comparison with a second-order FSM. A secondary concern regarding sentence boundary detection in mathematical reasoning tasks was also acknowledged, adding welcomed transparency to the methodology. While the false boundary rate remains relatively high, the method demonstrates sufficient robustness and yields meaningful performance overall. In light of the authors' thorough responses, I have decided to raise my score.

**Key Questions For Authors:**

Q1.

The six-state taxonomy is central to the entire framework, yet its completeness and domain coverage remain empirically unverified. Could the authors provide a quantitative analysis of annotation coverage or failure case such as the proportion of sentences that receive ambiguous (or unreasonable) state assignments, and discuss how the framework behaves when a reasoning step does not fit cleanly into any of the six predefined states? Additionally, are there plans or preliminary results suggesting how the taxonomy might be extended or adapted for domains beyond mathematics and scientific QA, such as code generation or open-ended reasoning?

Q2.

The FSM relies on a first-order Markov assumption, conditioning each transition solely on the immediately preceding state. However, reasoning behaviors such as backtracking or uncertainty estimation are likely influenced by context accumulated over multiple prior steps. Have the authors conducted any empirical analysis to assess how frequently this assumption is violated, and whether such violations measurably degrade the quality of the Transition Advantage Matrix or the effectiveness of steering? A simple analysis such as comparing first-order vs second-order transition statistics would substantially strengthen the soundness of the framework.

Q3.

By systematically steering reasoning trajectories toward high-advantage transitions, I guess the proposed method may contain a potential risk of reducing the diversity of generated reasoning paths. Similarly, injecting orthogonal activation vectors at sentence boundaries could introduce subtle degradations in output fluency or factual coherence. Could the authors provide any empirical evaluation of these potential side effects, such as measurements of output diversity across steered versus unsteered generations, or human evaluations of reasoning fluency, or perplexity? This would substantially clarify the practical trade-offs of deploying the method.

Q4.

The transition matrices are constructed from reasoning trajectories generated by each target model individually. While straight forward, It is unclear whether the captured transition dynamics are model-specific or reflecting more universal and general reasoning patterns. Could the authors provide a more systematic analysis of cross-model transferability of the advantage matrix, by evaluating whether a matrix derived from one model can meaningfully improve another model's performance? This would have important implications for the scalability and generalizability of the framework.

**Limitations:**

yes

**Strengths And Weaknesses:**

### Strengths

- The technical pipeline is well-constructed. The FSM formalization provides a clean, interpretable abstraction of reasoning trajectories. Also, each pipeline design, such as Q-value iteration or difference-of-means for extracting steering vectors is well established and clearly guided. The inter-annotator agreement validation (Cohen's Kappa = 0.89) also gives more reliability to the proposed pipeline.

- Figure 2 provides a useful visual overview of the end-to-end framework, and Figure 3 offers compelling qualitative examples of steering correcting flawed reasoning trajectories. The paper is organized logically, and the distinction between three steering policies, Greedy, Weighted, Q-Value, is clearly articulated.

- The paper makes a timely and valuable contribution by introducing a new analytical framework for understanding the internal reasoning dynamics of LRMs, and draws attention to a under-explored problem of the lack of global control over CoT trajectories. The paper proposes a practical inference-time intervention method and add interpretability to the the reasoning process. The insight that high-level cognitive transitions is valuable and potentially influential for future inference-time control research.

### Weakness

- The conceptual concern is the Markov assumption underlying the FSM. The transition probability $P(q_j | q_i)$ conditions only on the immediately preceding state, which might be unrealistic in practice. Behaviors such as uncertainty estimation or backtracking often arise from errors or inconsistencies accumulated across multiple prior reasoning steps, rather than from the single preceding state. While the authors acknowledge this limitation briefly, the paper would benefit from an empirical analysis of how frequently the Markov assumption is violated in practice. A simple analysis such as comparing first-order vs second-order transition statistics would substantially strengthen the soundness of the framework.

- The grounding of the six-state taxonomy deserves more thorough justification. Although the connection to Polya's four-step framework is noted, the paper does not sufficiently demonstrate that the proposed states are exhaustive or that they remain appropriate for reasoning trajectories produced by stronger future models.  Without stronger evidence of coverage, the method should re-design all finite through an iterative process of generation, observation, and state definition. An analysis of failure cases, particularly sentences that receive ambiguous or contested state annotations, would be practically valuable for researchers and practitioners who wish to apply this framework to new domains. Additionally, grounding the taxonomy more rigorously in relevant psychological or cognitive science literature would further strengthen its theoretical motivation.

- Also, the explanation of sentence boundary detection via punctuation tokens (".", "?", "!") raises a practical concern. Punctuation is frequently ambiguous in reasoning context. For example, decimal points (e.g., "1.3"), repeated punctuation (e.g., "!!"), or symbols embedded in code or equations can all be misinterpreted as sentence boundaries. The paper acknowledges this in the limitations appendix but does not report how often boundary detection fails in practice or what the fallback behavior is.

- While the authors describe their method as training-free and the activation steering itself does not require fine-tuning the underlying reasoning model, the proposed framework nonetheless relies on a dedicated training dataset and held-out development set for constructing the Transition Advantage Matrix and extracting steering vectors. Furthermore, two neural network classifiers, a current-state classifier and a next-state classifier, are trained as part of the pipeline. The claim of being "training-free" may be somewhat misleading and risks overstating the method's accessibility. The authors are encouraged to more carefully qualify this characterization.

### Minor issues

- There is a notation issue in the transition probability formula: the summation index $i=1$ to $N$ conflicts with the subscript $q_i$ in the denominator, creating ambiguity. Similarly, the definition of $D^+_{u→v}$ may contain a subscript error, $(u_k → v*{k+1})$ rather than $(u_k → v_k)$.

- Figure 2, while conceptually informative, is dense and would benefit from being enlarged.

- The QWEN MATH-500 weighted steering token count of 418.20 appears to be a typo (likely 4182.0)

---

> ### Author Rebuttal · Authors · 2026-03-31
>
> Thank you for your thoughtful comments and insightful suggestions. Please see our responses below.
>
> **Note: References are in our response to *Reviewer URnh* to optimize space.**
>
> ---
>
> **Q1) Justification and Generalizability of the State Taxonomy**: Besides Polya's four-step model (1945), our six-state taxonomy connects to another well-established cognitive framework – Schoenfeld's Episode Theory [1], which identifies six foundational episodes (Read, Analyze, Plan, Implement, Explore, Verify) as necessary and sufficient for complex problem-solving.
>
> Our taxonomy, based on empirical observations of the chains of thought, adapts this classical architecture to the context of LRMs by mapping Initialization to Read/Analyze, Deduction to Implementation, Augmentation to Plan/Explore, and Closure to Verify. We additionally incorporate Uncertainty and Backtracking to operationalize metacognitive monitoring and control functions (Nelson & Narens [2]), critical for capturing the non-linear, self-corrective behaviors unique to LRMs.
>
> Exploring the presence of additional finer-grained states is worthwhile, yet a more granular model risks descriptive redundancy and lower classification accuracy, while a coarser one (e.g., a simple linear Chain-of-Thought) would miss the strategic meta-level transitions that distinguish advanced reasoning from basic next-token prediction. This grounding will be included in the final version.
>
> ---
>
> **Q2) First-order Markov Assumption**: Please see our response under **Q1** of **Reviewer URnh**.
>
> ---
>
> **Q3) Practicality of Sentence Boundary Detection**: Sentence boundaries are well-motivated intervention points: Bogdan et al. [3] show sentence-level analysis best captures distinct reasoning steps, while Chauhan et al. [4] demonstrate punctuation tokens act as semantic summarization boundaries in LLMs. The inability to distinguish sentence-ending punctuation from decimals/equation symbols during autoregressive generation is an inherent constraint of inference-time methods operating at sentence granularity, not specific to our approach. We acknowledge robust boundary detection (e.g., lightweight segmentation models) as valuable future work.
>
> ---
>
> **Q4) Training-Free Framing Clarification**: Training-free refers to no weight updates to the LRM weights during inference, consistent with the activation steering literature [5-7]. We follow the offline-extract/online-steer setup of [8, 9]. Offline computation involving advantage matrix, vector extraction is lightweight and one-time: no gradients, no backpropagation. The two classifiers are tiny MLPs (512-dim, 2-layer) on frozen representations – negligible vs. even a 4B LRM. But thank you for the feedback, we will revise the paper to explicitly distinguish LRM-training-free inference from the offline preprocessing pipeline.
>
> ---
>
> **Q5) Annotation Failure Cases**: Please see our responses under **Q8** and **Q1** of **Reviewer hYyH**.
>
> ---
>
> **Q6) Diversity and Fluency Evaluation**: We conducted an empirical evaluation using 100 random problems from MATH-500 with QWEN, we ran 5 seeds per problem under both default and Q-Value steering conditions. We measured diversity via Self-BLEU and distinct-n (n=2), and fluency via perplexity (Qwen2.5-3B as reference LM), reporting mean ± std:
>
> | Metric | Default | Steered |
> |---|---|---|
> | Self-BLEU ↓ | 0.728±0.068 | 0.727±0.064 |
> | Distinct-2 ↑ | 0.373±0.060 | 0.372±0.057 |
> | Perplexity ↓ | 2.28±0.35 | 2.29±0.35 |
>
> The results are nearly identical across all metrics, indicating no clear loss of diversity or fluency under Q-Value steering. We plan to include complete results across all settings in the final version.
>
> ---
>
> **Q7) Cross-model Transferability**: Cross-model transferability is partially addressed in Appendix C (Table 2) via prompt-based Cross-Model Guidance. We additionally ran cross-model activation steering on MATH-500 (GPT-L using QWENs advantage matrix). Results show Greedy achieves 81.80% accuracy (307.18 tokens, 17.06 avg steering), Weighted achieves 81.60% (296.00 tokens, 14.58 avg steering), and Q-Value achieves 82.80% (293.85 tokens, 5.37 avg steering).
>
> Cross-model Q-Value steering yields 82.80% vs. 79.00% baseline. The higher intervention count is expected, as a transferred matrix is less calibrated and requires more corrections, while token counts remain stable. Together, prompt-based and steering-based results suggest cross-model transition dynamics are partially universal and partially model-specific. A full cross-model analysis across all datasets will be included in the final version.
>
> ---
>
> **Suggestion**: We will fix the transition probability formula by replacing the trajectory summation index with **n**. The ${D}^{+}\_{u \to v}$ formulation  is correct. We will add a clarification explicitly stating that **k** indexes sentence position and $u_{k}, v_{k}$ denote the source and target states at that position. We will also enlarge Figure 2 and fix the typo in the token count.

---

> > ### Author Rebuttal · Reviewer_RUUK · 2026-04-03
> >
> > Thank you for the detailed rebuttal, most of my concerns have been well addressed.
> >
> > Regarding on the first-order markov assumption, the sign agreement and correlation between first- and second-order matrices feels somewhat expected, as local state consistency is likely to hold between sequential orders. A more direct justification would be an empirical comparison of benchmark performance under first- vs. second-order models. Given the lightweight nature of the proposed method, I believe providing the final performance results would strengthen the justification of the method design a lot.
> >
> > Also, sentence boundaries remain a known source of noise, particularly in reasoning contexts. For example, in the process reward modeling, the community has similarly moved toward token-level granularity for this reason [1]. Though I understand the inability of sentence boundary is not specific to the proposed approach, as one of the early works on sentence-level reasoning trace steering, providing concrete failure cases of this granularity on reasoning domain would meaningfully strengthen the paper's contribution, even if a full solution is left to future work.
> >
> > [1] Cui, Ganqu, et al. "Process reinforcement through implicit rewards." arXiv preprint (2025).

---

> > > ### Author Response · Authors · 2026-04-06
> > >
> > > **Q1)** Thank you for this excellent suggestion to improve the paper. Within the discussion window, we implemented second-order FSM steering across all three steering methods on the MATH-500 dataset.
> > >
> > > For Greedy and Weighted, the steering policy uses second order advantage matrix R²[(prev, curr, next)] with first order matrix R¹ as fallback for structurally absent pairs.
> > >
> > > For Q-Value, we tested three variants:
> > >
> > > (1) *Fallback*: immediate reward from R² with future value from R¹ based Q-table,
> > >
> > > (2) *Interpolation*: blended λ·R²+(1−λ)·R¹ where λ = min(count/threshold, 1), here count = number of distinct prev states for which R²[(prev, curr, next)] is observed and threshold = 4, and
> > >
> > > (3) *Full Second-Order*: immediate reward from R² with R²-mean (mean of R² over observed prev states, R¹ fallback for absent pairs) Q-table. In all variants, previous state is tracked at inference with no new classifier needed.
> > >
> > > ---
> > >
> > > **Table 1.** Results of Greedy and Weighted steering. Here, Acc = Accuracy (%), AT = Average number of tokens, AS = Average number of steering actions, GFO = accuracy gain of second-order over first-order.
> > >
> > > | Model | Acc | AT | AS | GFO | Acc | AT | AS | GFO |
> > > |---|---|---|---|---|---|---|---|---|
> > > | | | **Greedy** | | | | **Weighted** | | |
> > > | GPT-L | 81.80 | 328.74 | 17.32 | +0.60 | 79.80 | 291.50 | 14.24 | -2.60 |
> > > | GPT-M | 85.40 | 1109.70 | 39.46 | 0.00 | 85.80 | 1251.07 | 40.54 | -0.80 |
> > > | PHI | 90.00 | 1379.74 | 38.87 | +0.80 | 89.40 | 1388.83 | 37.87 | -1.40 |
> > > | QWEN | 91.00 | 4159.25 | 82.49 | 0.00 | 90.60 | 4188.78 | 45.22 | -0.40 |
> > >
> > > ---
> > >
> > > **Table 2.** Q-Value steering: Fallback and Interpolation variants.
> > >
> > > | Model | Acc | AT | AS | GFO | Acc | AT | AS | GFO |
> > > |---|---|---|---|---|---|---|---|---|
> > > | | | **Fallback** | | | | **Interpolation** | | |
> > > | GPT-L | 82.80 | 302.10 | 6.37 | -0.40 | 82.00 | 315.32 | 6.72 | -1.20 |
> > > | GPT-M | 86.00 | 1248.92 | 13.73 | -1.00 | 85.40 | 1191.57 | 14.39 | -1.60 |
> > > | PHI | 89.40 | 1468.57 | 4.66 | -0.20 | 89.40 | 1466.54 | 4.47 | -0.20 |
> > > | QWEN | 91.20 | 4226.58 | 17.95 | -0.20 | 91.00 | 4246.81 | 17.86 | -0.40 |
> > >
> > > ---
> > >
> > > **Table 3.** Q-Value steering: Full second-order variant
> > >
> > > | Model | Acc | AT | AS | GFO |
> > > |---|---|---|---|---|
> > > | GPT-L | 82.60 | 315.19 | 7.71 | -0.60 |
> > > | GPT-M | 85.60 | 1205.04 | 9.94 | -1.40 |
> > > | PHI | 88.80 | 1485.73 | 4.02 | -0.80 |
> > > | QWEN | 91.40 | 4251.58 | 17.01 | 0.00 |
> > >
> > > ---
> > >
> > > The empirical comparison across five steering configurations and four models consistently shows that second-order modeling yields little or no accuracy benefit over the first-order Markov design choice. If the second-order information were genuinely capturing additional structure beyond first-order, we would expect consistent positive GFO values. Instead, GFO is slightly positive only for Greedy (GPT-L: +0.60, PHI: +0.80) and largely neutral or slightly negative elsewhere.
> > >
> > > However, thank you for this valuable suggestion, which has strengthened our empirical validation. We will include second-order steering results across all benchmarks in the final version.  We believe that this direction remains promising; for example, we may find different advantage matrices that correspond to different stages of the reasoning (early vs. late) or that are based on the nature of the reasoning task, and hope to explore it more in subsequent work.
> > >
> > > ---
> > >
> > > **Q2)** Thank you for the suggestion. During the discussion window, we have conducted diagnostic experiments on the MATH 500 benchmark.
> > >
> > > Across all four models, the overall false boundary rates are 84.73% (GPT-L), 84.90% (GPT-M), 88.70% (PHI), and 90.23% (QWEN). False boundary triggers are mostly caused by ellipsis tokens ("..."), accounting for 99.6% (GPT-L), 99.4% (GPT-M), and 99.6% (PHI) of false triggers, with QWEN at 93.2%. The remaining cases include relatively rare cases such as repeated exclamation marks, decimal points, in-formula periods, factorials and abbreviations. It is perhaps possible to augment the classifier to handle the most common of these cases.
> > >
> > > The confidence gate largely mitigates this noise. Our gating mechanism (classifier confidence >= 90%) suppresses the majority of false boundary triggers: 35.6% (GPT-L), 56.0% (GPT-M), 80.0% (PHI), and 74.3% (QWEN). Slip-through rates (false boundaries that actually cause steering) remain low at 6.7%, 0.1%, 7.3%, and 18.9% respectively, demonstrating effective practical mitigation.
> > >
> > > The confidence gap confirms discriminability for most models. For GPT-L and QWEN, false boundaries elicit lower classifier confidence than true ones (gaps of +0.057 and +0.007). For GPT-M and PHI, the gap is slightly reversed (−0.015 and −0.004), meaning false boundaries elicit marginally higher classifier confidence, indicating that the confidence gate alone is insufficient in these cases and motivating future token-level granularity refinements as pursued in [1] which you mentioned.
> > >
> > > We will incorporate additional failure case statistics and breakdowns as an appendix in the final version.

---

### Decision · Program_Chairs · 2026-04-30

**Decision:**

Accept (spotlight)

**Comment:**

This paper proposes a novel framework to model and control the reasoning process of large reasoning models (LRMs) through a finite state machine (FSM) abstraction. The central idea is to map long Chain-of-Thought (CoT) trajectories into sequences of six high-level cognitive states (e.g., deduction, backtracking, uncertainty), enabling a structured analysis of reasoning dynamics.  ￼

The study proceeds to address the concept of representing reasoning as transitions over discrete cognitive states, providing a compact and interpretable view of otherwise opaque token-level trajectories. The authors attempt to analyze a notable aspect of reasoning behavior by distinguishing transition patterns between successful and failed trajectories via a Transition Advantage Matrix.

Building upon this abstraction, the paper introduces a planning-based inference-time control mechanism using Q-value iteration, which selectively steers the model’s hidden activations toward more favorable reasoning transitions. Experimental results across multiple benchmarks (AIME25, MATH-500, GSM8K, GPQA) demonstrate consistent accuracy improvements with significantly fewer interventions.

Basically, the idea of this paper is novel, and the experiments are convincing. All reviewers give positive scores on this paper. Though there are some weaknesses, such as the Markov Assumption may not hold in real-world applications, but in my view, it does impact that this paper is a valuable work to help understand the reasoning ability of LLMs.